# What Happens During the Loss Plateau? Understanding Abrupt Learning in Transformers

**Pulkit Gopalani**
University of Michigan, Ann Arbor
`gopalani@umich.edu`

**Wei Hu**
University of Michigan, Ann Arbor
`vvh@umich.edu`

## Abstract

Training Transformers on algorithmic tasks frequently demonstrates an intriguing *abrupt learning* phenomenon: an extended performance plateau followed by a sudden, sharp improvement. This work investigates the underlying mechanisms for such dynamics, primarily in shallow Transformers. We reveal that during the plateau, the model often develops an interpretable *partial solution* while simultaneously exhibiting a strong *repetition bias* in their outputs. This output degeneracy is accompanied by *internal representation collapse*, where hidden states across different tokens become nearly parallel. We further identify the slow learning of optimal attention maps as a key bottleneck. Hidden progress in attention configuration during the plateau precedes the eventual rapid convergence, and directly intervening on attention significantly alters plateau duration and the severity of repetition bias and representational collapse. We validate that these identified phenomena—repetition bias and representation collapse—are not artifacts of toy setups but also manifest in the early pre-training stage of large language models like Pythia and OLMo.

## 1 Introduction

Training Transformers on mathematical or algorithmic tasks often exhibits an intriguing "abrupt learning" phenomenon in their training dynamics, where the model's performance plateaus at a suboptimal level for an extended period before suddenly and rapidly converging to the optimal solution [3, 36, 44, 47, 54] (Figures 1 and 2). This is often considered an example of the broader phenomenon of "emergence," where model capabilities appear to arise discontinuously and unpredictably with increasing amount of parameters, training data, or training steps [49]. Understanding these sharp phase transitions in learning trajectories is crucial for gaining deeper insights into how Transformer models learn and develop their sophisticated capabilities.

Despite recent progress in understanding such abrupt learning dynamics in Transformers for specific tasks like in-context learning [10, 15, 41, 44, 47, 54], parity learning [3], Markov chains [15], grammar learning [9, 32], and matrix completion [18], a unifying account of the model evolution during loss plateau is still missing. Further, many of these works require specific assumptions on model and data that limit their generality.

The goal of this paper is to uncover universal characteristics and underlying mechanisms that define these training dynamics that are broadly applicable to a wide range of setups and tasks. Specifically, what common patterns manifest in the model's input-output behavior and internal representations during the extended plateau phase, and what critical changes precede the sudden shift towards higher performance? Is there hidden progress accumulating beneath the surface of the loss plateau? Answering these questions is pivotal for building a comprehensive picture of the nature of phase transitions in Transformer training dynamics.

39th Conference on Neural Information Processing Systems (NeurIPS 2025).

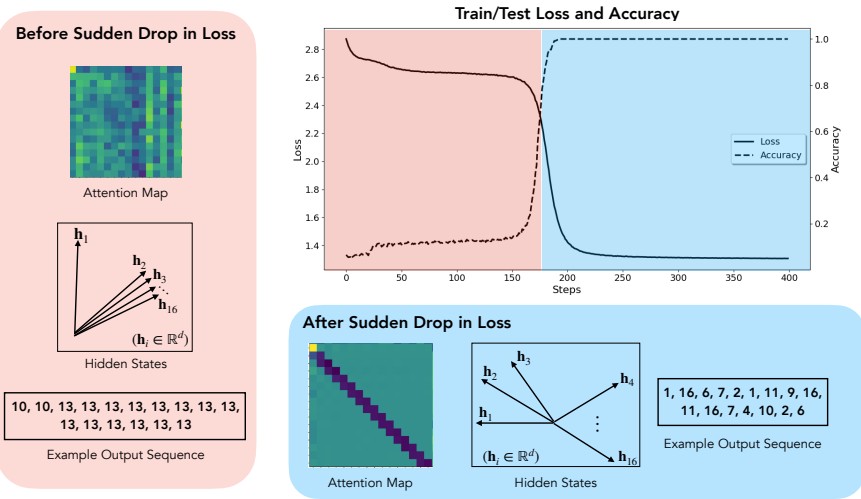

Figure 1: **Abrupt learning and related characteristics.** Training a shallow Transformer on algorithmic tasks like moving-window-sum exhibits an *abrupt learning* curve: performance plateaus for an extended number of steps, before suddenly and sharply improving to optimum. Before the sudden drop in loss, the attention map cannot be interpreted easily, whereas the post-sudden-drop attention map is clearly interpretable w.r.t. the task. Furthermore, the model exhibits degenerate patterns before the sudden drop, including output repetitions and collapse of its hidden representations.

To investigate these questions systematically and within a controlled setting, we focus on training small, shallow Transformers (typically 1 or 2 layers) on a suite of simple algorithmic tasks. The reduced model size allows for more tractable analysis and clearer interpretation of internal model mechanisms, avoiding the obfuscation that can arise from the interplay of countless factors in large models. Furthermore, algorithmic tasks such as moving-window-sum, prefix-sum, and multi-digit addition (as detailed later) have well-defined optimal solutions, thus allowing us to precisely measure the model's progress against a known ground truth and to readily interpret which aspects of the problem the model is succeeding on at different training stages. Interestingly, we will demonstrate that key findings from these controlled small-scale studies extend to the pre-training dynamics of actual Large Language Models (LLMs).

**Our Contributions.** We identify implicit biases that underlie the early plateau period of Transformer training: the model learns a partial solution while being biased toward degenerate patterns in its outputs and internal representations. We further study the pivotal role of attention map learning in driving these phenomena and overcoming the performance plateau. See Figure 1 for an overview of our findings. Our specific contributions are:

- **Partial solutions during plateau:** We show that during the initial loss plateau, the model often learns a **partial solution**, which correctly predicts a subset of easier tokens within a sequence—those that might be intuitively simpler to learn (e.g., copying the first element in a moving-window sum, or predicting the final carry-over in multi-digit addition)—while failing on more complex parts of the task. This pattern is observed across diverse algorithmic tasks (Table 1).

- **Repetition bias in outputs:** We identify a strong **repetition bias** during the plateau, where the model tends to output repetitive tokens. This bias can be quantified by metrics such as a direct count of repeated subsequent tokens or the entropy of the output token distribution. Such repetitions significantly increase during the early training steps and then markedly decrease as the performance starts to improve (Figure 2b).

- **Internal representation collapse:** The output repetition bias is accompanied by **representation collapse**, where hidden representations for different tokens become nearly parallel (e.g., cosine similarity often exceeds 0.9), indicating a degenerate representational geometry inside Transformers. Subsequently, this representational similarity drops significantly as the model's performance improves, signifying a diversification of internal representations (Figure 2b).

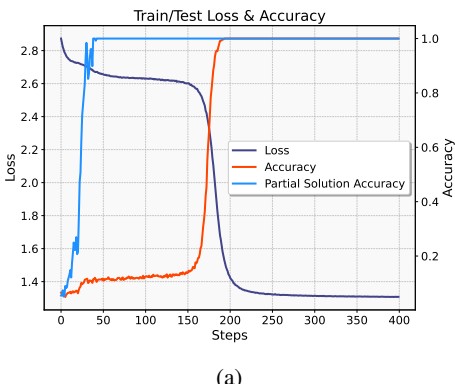
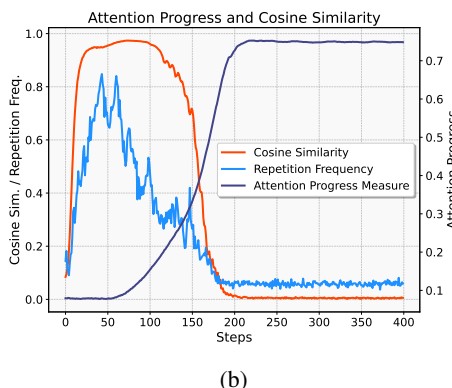

|(a)|(b)|

Figure 2: **Abrupt learning dynamics for the MWS task.** (a): Train/Test loss and Train/Test Accuracy (note that both train and test data metrics are near-identical in the online training setup, and thus we only report train metrics); (b): Attention Progress, Repetition Frequency, and Representation Cosine Similarity between hidden states. Increase in attention progress is gradual and happens before the sudden loss drop. Repetition frequency and representation cosine similarity rapidly increase at the beginning and decrease to low values later on.

- **Crucial role of attention map learning:** We find that gradual learning of the optimal attention pattern can commence during the loss plateau, before the sudden drop in loss (Figure 2b). By directly intervening on the attention map during training—for instance, by biasing the attention scores towards or away from the optimal configuration—we can observe tangible changes in the duration of loss plateau and the severity of degenerate behaviors like repetition bias and representation collapse.

- **Validation in LLMs:** Our identified phenomena of repetition bias and representation collapse are not limited to small Transformers on synthetic algorithmic tasks. We further validate their occurrence in the early pre-training phases of LLMs like Pythia and OLMo, suggesting these are general characteristics of Transformer training dynamics.

## 2 Setup, Abrupt Learning, and Attention Map

We mainly present results for the moving-window-sum task in the main text, which we define below; we also validate our findings on various other algorithmic tasks like multi-digit addition, permutations, histogram, prefix-sum, etc., in Appendix B.

**Data.** The moving-window-sum (MWS) task involves computing the sliding-window sum (modulo $p$) of a length-$n$ sequence over windows of size 2; that is, sequences in MWS are

$$x_1, x_2, \ldots, x_n, \text{SEP}, y_1, y_2, \ldots, y_n$$

$$y_i = \begin{cases} x_1 & i = 1 \\ (x_{i-1} + x_i) \bmod p & i \geq 2 \end{cases}$$

Here, $x_1, \ldots, x_n$ are the input sequence, SEP is a separator token, and the task is to complete the sequence with outputs $y_1, \ldots, y_n$. In the experiments in the main paper, we use $n = 16$, $p = 17$, $x_i \sim \text{Unif}\{1, 2, \ldots, 16\}$ and $\text{SEP} = 17$. We denote the full vocabulary $V := \{0, 1, 2, \ldots, 17\}$, and directly use these integers as token IDs when generating token embeddings for input to the model.

**Model Architecture.** We use a 1-layer, 1-head Transformer with causal masking and linear attention. This simple architecture can already solve the MWS task to perfect accuracy. Formally, for a sequence of tokens $(s_1, \ldots, s_L)$, the Transformer output is,

$$\text{TF}_\theta(s_1, s_2, \ldots, s_L) = \text{LM} \circ (\text{Id} + \text{MLP}) \circ (\text{Id} + \text{Attn}) \circ \text{Embed}(s_1, s_2, \ldots, s_L) \quad (1)$$

where Embed outputs sum of token and absolute positional embeddings $h_i \in \mathbb{R}^d$, and Attn denotes the causal-linear-Attention operation that combines tokens such that output at $i^{th}$ position is,

$$[\text{Attn}(h_1, h_2, \ldots, h_L)]_i = W_O \left( \sum_{j=1}^{i} (h_j^\top W_K^\top W_Q h_i) W_V h_j \right); W_O, W_K, W_Q, W_V \in \mathbb{R}^{d \times d} \quad (2)$$

MLP denotes the 2-layer neural net $h_i \mapsto W_2(\sigma(W_1 h_i))$ for $W_2 \in \mathbb{R}^{d \times 4d}, W_1 \in \mathbb{R}^{4d \times d}$, and $\sigma$ the GELU activation. LM is a linear layer that maps the hidden state $h_i \in \mathbb{R}^d$ to logits $v_i \in \mathbb{R}^{|V|}$. Note that all linear maps above implicitly include a bias term, and we use pre-LayerNorm so that before the Attn, MLP, and LM, a LayerNorm operation is applied to the hidden states $h_i$. For generating sequences, we use greedy decoding i.e. output token is determined by the maximum logit over the vocabulary (please see Appendix A for more implementation details). We use linear attention to avoid small gradient issues from softmax function being a contributing factor toward abrupt learning, as argued in [20]. We also show similar results on softmax attention, multi-layer / multi-head models, and models with varying $d$ in Appendix F.

**Training.** The model is trained to minimize the standard next-token-prediction cross-entropy loss over the full sequence i.e. $(x_1, \ldots, x_n, \text{SEP}, y_1, \ldots y_n)$ for the MWS task. We evaluate accuracy over the output portion of the sequence, i.e., $y_1, \ldots, y_n$, averaged over these $n$ positions. We use the Adam optimizer with a constant learning rate $10^{-4}$ and no weight decay. The training is conducted in an online / single-epoch fashion, where a new batch of 256 training samples is drawn from the data distribution at each training step. Note that in this setup, the training and test losses essentially coincide. For completeness, we verify in Appendix G that our reported phenomena occur when using different optimization algorithms (SGD, Muon [25]) or optimization hyperparameters (learning rate, LR schedules, batch size, initialization scale, weight decay) as well, and are not merely an artifact of a specific training setup. Code for experiments is available at `github.com/pulkitgopalani/tf-loss-plateau`.

**Abrupt Learning.** Following the training procedure described above will result in a characteristic *abrupt learning* curve, where the training/test loss is stuck at some sub-optimal value for a significant number of steps, before suddenly and rapidly decreasing to its optimal value (Figure 2a). This drop in loss is accompanied by a similarly rapid increase in accuracy, indicating that the optimal solution is learned abruptly.

**Attention Map.** We analyze the attention map at different points during training. We find that the attention map shows a sparse, interpretable pattern after the sudden loss drop, while no such pattern is shown before the sudden drop (Figure 1). For the MWS task, this optimal attention pattern corresponds to each output token $y_i$ attending only to the input tokens relevant to its computation, i.e., attending to $x_1$ for $y_1$, and to $x_i, x_{i-1}$ for $y_i, i \geq 2$. We further use an *Attention Progress Measure (APM)* to record the progress of the attention map toward its optimal pattern during training, defined as

$$\text{APM} := \frac{\sum_{(i,j) \in \Omega} |A_{ij}|}{\sum_{(i,j)} |A_{ij}|}, \quad (3)$$

where $A_{ij}$ denotes the attention score allocated to the $j^{th}$ token when computing output at the $i^{th}$ position in the sequence, and $\Omega$ is the set of position pairs in the optimal attention map. This measure is defined with absolute values due to our choice of linear attention so that $A_{ij}$ could be positive or negative. In experiments, we calculate APM averaged over a random batch of sequences. The APM sets $\Omega$ for all relevant tasks in this paper are defined in Table 2.

Figure 2b shows that the APM monotonically increases from near $0$ to near $0.8$ during training, and its increase is more gradual than the loss/accuracy dynamics. In particular, APM already increases to a nontrivial value during the loss plateau and before the sudden loss drop.

## 3 Implicit Biases in the Early Phase of Training

In this section, we characterize several key manifestations of the implicit biases in the early phase of Transformer training. These patterns robustly co-occur with the loss plateau, and provide intuitive indicators that the model is getting stuck at a degenerate state during the plateau.

**Partial Solution.** During the loss plateau, the model often has already learned to implement a *partial solution* to the task. This means it correctly predicts a subset of the output tokens, typically those corresponding to an intuitively simpler part of the problem, while failing on the more complex parts. For instance, in the MWS task, the model quickly learns to predict the first output token $y_1$ correctly (see Figure 2a for the first-token accuracy), as it is simply a copy of the first input token $x_1$, while the overall loss remains high and accuracy on subsequent tokens is poor. This ability to solve easier sub-components of the task early on is observed across various algorithmic problems (see Table 1 in Appendix B).

**Repetition Bias.** Concurrent with learning the partial solution, the model's outputs during the initial phase of training display a strong *repetition bias*, which refers to a tendency of the model to generate repetitive tokens of the form $x, x, x, \ldots$. One way to quantify such repetitions is to simply count the frequency of output tokens that equal the next one: for output sequence $y_1, y_2, \ldots, y_n$, define its repetition frequency as,

$$\rho := \frac{1}{n-1} \sum_{i=1}^{n-1} \mathbf{1}[y_i = y_{i+1}]. \tag{4}$$

We observe that $\rho$ increases rapidly during the early phase of training, when the optimal attention map has not been learned yet (Figure 2b). Note that this frequency is small at initialization, and grows rapidly in the first $\approx 50$ steps to $\approx 0.8$, indicating that it is an *implicit bias coming from gradient-based training*.

**Representation Collapse.** Motivated by the frequent repetitions in model outputs, we further study the relation between the hidden representations at different output positions. We find a strong *representation collapse* phenomenon—these representations become nearly parallel in the early phase of training (except for the first output position which is correctly predicted in the partial solution). We measure the pairwise cosine similarity between hidden representations at positions $i, j$ in the output,

$$\text{COS}_{i,j} := \frac{\langle \mathbf{h}_i, \mathbf{h}_j \rangle}{\|\mathbf{h}_i\|\|\mathbf{h}_j\|} \tag{5}$$

where $\mathbf{h}_i \in \mathbb{R}^d$ is the hidden state at position $i$ in the sequence (this quantity is averaged over a random batch of sequences). We find that in the early phase of training, there is a rapid increase in $\text{COS}_{i,j}$—averaged over all output positions $i, j$ except the first position, this quantity increases to $\approx 0.95$ (Figure 2b). We emphasize that similar to repetitions, representation collapse is not present at initialization and only appears after a few steps of training. This is in contrast to the *rank collapse* phenomenon for deep softmax-attention Transformers [2] that occurs at initialization. Also, while we focus on the final-layer representation (before the LM layer) in the main text, we show in Figure 42 that representation collapse happens in all intermediate layers to varying degrees.

## 4 The Role of Learning Attention

Observe that though the loss dynamics are abrupt, attention progress measure as well as repetitions and representation collapse are not (Figure 2b); that is, even when the loss is barely decreasing (between steps 50 and 150), attention progress measure notably increases, accompanied by a decrease in repetition frequency and representation collapse. Via training-time interventions, this section shows that learning the attention map plays a crucial role in shaping the loss plateau as well as repetitions and representation collapse.

**Representation Collapse Occurs After the Attention Layer.** We start by verifying whether the attention layer is responsible for representation collapse during the early phase of training. To this end, we plot the cosine similarity of the residual stream for output tokens just before and after the attention layer. Formally, let the residual stream before attention layer (i.e., token + positional embeddings) be $\mathbf{h}_i \in \mathbb{R}^d$, and the residual stream after attention layer be $\mathbf{h}'_i \in \mathbb{R}^d$, we measure the norm and pairwise cosine similarity for $\mathbf{h}_i$ and $\mathbf{h}'_i$ in Figure 41.

We find that in the early phase of training, the cosine similarity between different positions in the post-attention residual stream representations approaches 1.0 rapidly, which is not the case for pre-attention. Furthermore, the norm of $\mathbf{h}'_i$ grows rapidly in this phase, while the norm of $\mathbf{h}_i$ remains

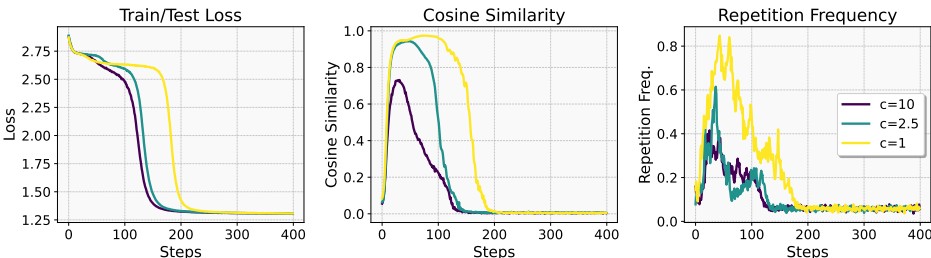

Figure 3: **Biasing attention map by** $c > 1$. We find that multiplicative biasing the attention map towards more weight to optimal positions leads to faster convergence, accompanied by less repetitions and average cosine similarity.

near-constant. *Hence, in the residual stream, representation collapse occurs after the attention layer during the early phase of training.*

**Biasing the Attention Map.** To study the role of attention map, we slightly modify the training process starting at different time points in training, biasing it towards (or away from) the optimal attention map to check if repetitions, representation collapse, and loss plateau are reduced (resp. amplified). We do the following: For each step in training after $t_0$, for attention map $A \in \mathbb{R}^{(L-1)\times(L-1)}$, we use the mask $M \in \mathbb{R}^{(L-1)\times(L-1)}$, $M_{ij} = c$ for $(i,j) \in \Omega$, $M_{ij} = 1$ otherwise, except for $i = 1$ (since that is a partial solution and converges early in training). Then, we use the modified attention map $A \odot M$ (Hadamard product) for training and inference. Hence, for $c > 1$, this implies biasing the model towards the final (optimal) attention map, whereas for $0 < c < 1$, this implies biasing the model away from the optimal attention map.

We find that, for $c > 1$ and various values of $t_0$, such a scaling leads to lower average cosine similarity between hidden states, lower frequency of repetitions, and faster convergence (Figures 3 and 43). Whereas, for $0 < c < 1$, we find the opposite: the model is in representation collapse state for a longer time and converges later compared to the non-scaled ($c = 1$) case, while the repetition frequency remains large throughout the plateau (Figure 4).

For example, for $t_0 = 0, c = 10$, i.e. scaling $10\times$ from the start of training, we find that the peak cosine similarity attained during training is $\approx 0.6$, much smaller than the $\approx 0.95$ attained for $c = 1$, and further the peak for $c = 10$ is for negligible duration compared to that for $c = 1$. Later values of $t_0 = 25, 50, 75$ show similar results wherein the cosine similarity drops immediately on the above biasing operation, followed by lower repetition frequency and convergence to optimal solution (Figure 43). On the other hand, for $t_0 = 0, c = 0.2, 0.5$, the model takes much longer to converge and is in representation collapse / large repetition frequency state for much longer. This is in line with our expectation that lower attention map values for the optimal positions lead to slower learning and prolonged representation collapse.

*Hence, learning the optimal attention map has a direct effect on shaping the loss dynamics as well as repetitions and representation collapse.*

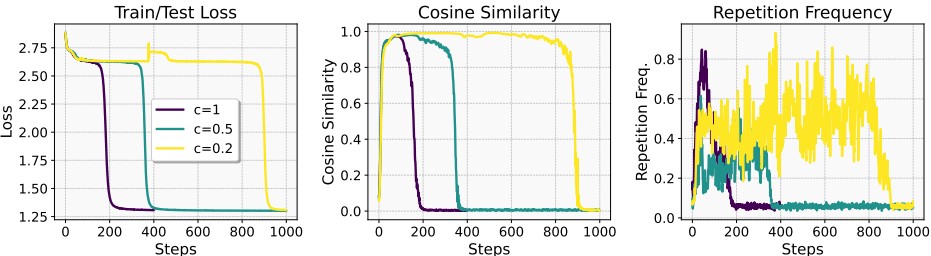

Figure 4: **Biasing attention map by** $c < 1$. We find that biasing the attention map to have lesser weight at optimal positions leads to slower convergence, and more representation collapse and repetitions.

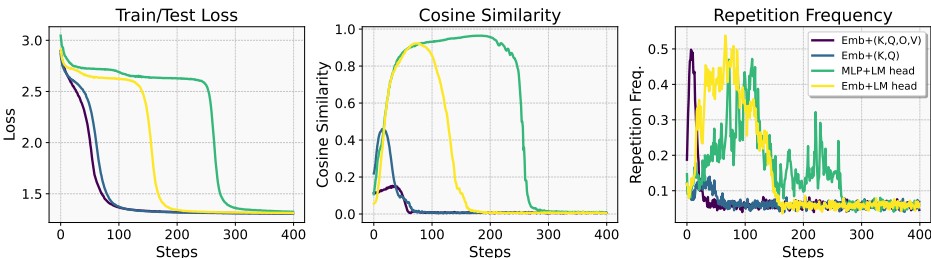

Figure 5: **Different optimal initializations and effect on training.** We find that fixing attention and embedding weights (i.e. attention map) to optimal value, and training other components leads to faster convergence and lesser representation collapse / repetitions. Similar effect does not hold for fixing optimal MLP or Embeddings. ($K, Q, O, V$ respectively denote the parameters $W_K, W_Q, W_O, W_V$.)

**Training with Optimal Attention Map v. Other Components**    In the first part of this test, we initialize with the optimal attention map by fixing embeddings, LayerNorm for attention layer and attention layer weights to their final values at the end of a normal training run, so that at initialization, the correct attention map is already available to subsequent layers. We re-train the subsequent non-fixed layers starting from random initialization. For the attention layer, we choose the set of parameters to initalize in 2 ways: (a) only Key, Query ($W_K, W_Q$) weights, and (b) All of Key, Query, Value, Output ($W_K, W_Q, W_V, W_O$) weights. We find that in both of these cases, learning only the subsequent layers (i.e. MLP, LM Head) take significantly shorter time than training the full model, without any significant representation collapse, repetitions or plateau in loss (Figure 5). Further, between (a) and (b), we find that additionally having $W_O, W_V$ layers initialized to optimal values slightly speeds up learning, and average cosine similarity goes up to approx $0.15$ instead of $\approx 0.45$ when only initializing $W_K, W_Q$ weights. This indicates that $W_O, W_V$ layers also play a non-trivial role in causing representation collapse.

On the other hand, fixing MLP or embeddings (together with LM head) to their final optimal values and re-training the other components does not qualitatively change the training dynamics from the full training case, i.e., a significant loss plateau, repetition bias, and representation collapse still occur (Figure 5).

*This result confirms that attention map is a major bottleneck that leads to early representation collapse and loss plateau, and that there is little benefit from having the optimal MLP or embeddings at initialization compared to attention map.*

Our results are similar to [44] that shows how various sub-circuits (incl. attention map) for in-context learning affect the loss plateau via training-time interventions, and [29] that shows the effectiveness of attention transfer from pretrained model for downstream task training in Vision Transformers.

**Tracking Progress in Hidden States**    To further track model progress during training, we probe the output of Attention layer before it is added to the residual stream (Equation (2)) using a 2-layer MLP probe with ReLU activation, and find that the accuracy for this probe increases earlier than the model accuracy. Please see Appendix C for more details.

**Searching for the Right Tokens**    The preceding results have demonstrated the important role of learning attention map (i.e., attending to the correct tokens) towards the plateau in training loss. Furthermore, once there is sufficient progress in the Attention map (visualized via APM in Figure 2b and output probing in Figure 20), the final model output converges rapidly to the correct solution. These observations conceptually align with abrupt learning observed when training 2–layer neural nets on a specific class of target functions [1], where the loss plateau is attributed to 'search' phase for aligning the first layer with the support of target function. In our setup, we hypothesize that a similar 'search' phenomenon could be occurring, where gradual increase in attention map weights at the required input tokens underlies the loss plateau.

Similar progress in model weights during loss plateau for a sinusoidal neuron trained on a sparse parity task was shown in [3, Fig. 3]. [38] discuss parallel search when training multi-head Transformers on a sparse parity task, and how the number of attention heads affects training dynamics.

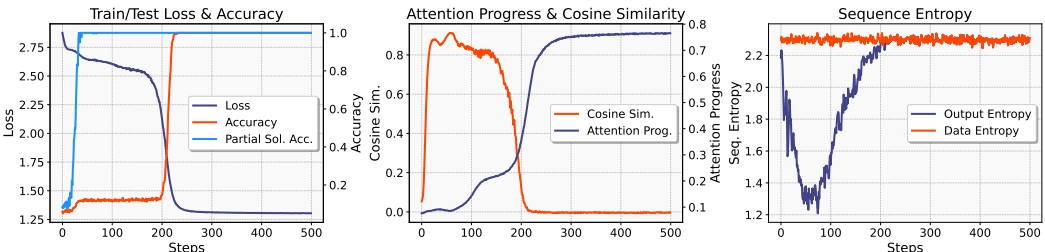

Figure 6: **Prefix sum task training dynamics.** While the usual contiguous repetitions do not occur for this task, an alternate form of repetition occurs in terms of having only a few distinct tokens in the output sequence. 'Sequence entropy' quantifies this repetition by measuring the entropy of the empirical distribution of tokens in a sequence, and averaging this entropy over a batch of sequences.

## 5 A Further Look at Repetition Bias

Having observed that Transformer models exhibit a strong repetition bias in the early phase of training, which co-occurs with the loss plateau, we now take a further look at this repetition bias and study how it might be affected by the amount of repetitions in the training sequences. We show that such bias still exists even when there is almost no repetition in the training data. Subsequently, we also show that learning simple, repetitive sequences is easier for Transformers, i.e., no loss plateau, indicating why the model might be biased towards repetitions in the early phase of training.

**Beyond Repetitions in Consecutive Tokens**    One hypothesis for the reason behind repetition bias is that the training data may consist of some repetitions, and the model may pick up these patterns and amplify them in the early phase of training. To investigate this, we consider a task with low repetitions in the training data. In particular, we consider the *prefix sum* task, where the outputs $y_1, \ldots, y_n$ are defined as $y_i = (\sum_{j=1}^{i} x_j) \bmod p$. Our choice of the input distribution ensures that there is no repetition in consecutive output positions (i.e., $y_i \neq y_{i+1}$ for all $i$). Indeed, training a Transformer on the prefix sum task does not result in a significant increase in the repetition frequency at any point in training, unlike the MWS task. Nevertheless, in the early training phase, we still observe that only a few tokens appear repeatedly in the model output though not contiguously as in the MWS task. Therefore, we consider an alternative measure of repetitions based on entropy: for an output sequence $y_1, y_2, \ldots, y_n$, we define

$$\text{SeqEnt}(y_1, \ldots, y_n) := \sum_{i=1}^{|V|} p_i \log(1/p_i); \quad p_i = \frac{|\{y_j = v_i, j \in [n]\}|}{n} \tag{6}$$

i.e. simply the entropy of the empirical distribution of tokens in the sequence. Intuitively, the entropy is lower if most probability mass is concentrated at a few tokens, and larger if the tokens are more uniformly distributed. We find that the model output entropy quickly goes to quite low values early in training compared to the entropy of ground-truth data (Figure 6), indicating that the model still has a form of repetition bias. Further, representation collapse still happens in the early phase, with the average cosine similarity going to $0.8$ during the plateau. *Hence, we find that repetition bias might take different forms depending on the task, but still robustly occurs in the early phase of training.*

**Repetitive Sequences are Easier to Learn**    We study what happens when training our model on data that has a lot of repetitions. We consider a simple task $\text{REPEAT}_1$ of the form $x_1, x_2, \ldots, x_n, \text{SEP}, y_1, y_2, \ldots, y_n$, where $y_i = x_1 \ \forall i$. Unlike other tasks, the loss curve for $\text{REPEAT}_1$ does not have any noticeable plateau, though the accuracy still shows a small plateau period (Figure 7). This observation indicates that such repetitive sequences are easier from an optimization perspective and hence likely "preferred" during the early stage of training. In fact, just one gradient step is sufficient to bring the average representation cosine similarity to $\approx 0.5$. We show similar results for other task variants ($\text{REPEAT}_2, \text{REPEAT}_4$) in Appendix D.

To further understand the early training phase model output, we define another metric $\alpha_1$ that measures to what extent the model simply outputs the same token for all output positions: $\alpha_1 = \frac{1}{n} \sum_{i=1}^{n} \mathbf{1}[y_i = y_1]$. Note that this is distinct from accuracy, in that the model might output the wrong

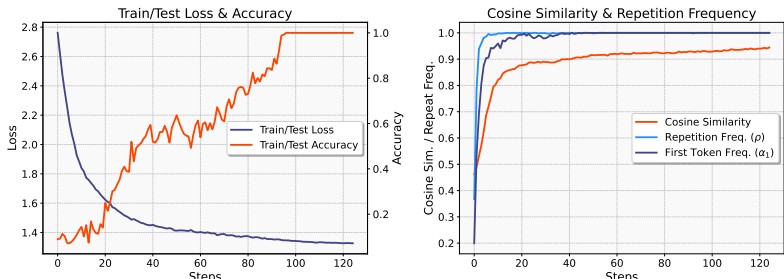

Figure 7: REPEAT$_1$ training dynamics.

$y_1$, however repeats $y_1$ at $y_2, \ldots, y_n$. We find that $\alpha_1$ rapidly increases to near perfect values ($> 0.9$) in the early phase of training, showing that the model tends to repeat the first token identically at most positions, even though the output token itself might be incorrect. *Hence, repetitive sequences appear to be inherently easier for the Transformer to learn, and this is likely the reason for repetition bias in the early phase of training.*

## 6 Repetition Bias and Representation Collapse in LLMs

Having established that the degenerate patterns of repetition bias and representation collapse are prevalent in small Transformers trained on algorithmic tasks, a crucial question is whether these phenomena also happen beyond toy settings, specifically during the early pre-training stages of LLMs. We verify that this is indeed the case, using early-stage checkpoints of open-source LLMs Pythia [6] and OLMo-2 [37].

For Pythia models with 14M, 1B, 1.4B, and 2.8B parameters, we find strong representation collapse in the early training steps in their last layer and repetition bias in the output sequence (Figure 8). Specifically, we randomly sample 100 questions from the test split of the AI2 ARC-Easy dataset [12] . For each question, we let the model generate 8 tokens, and compute the pairwise cosine similarity of the hidden states (see Appendix A for more implementation details). Figure 8 shows that at initialization, the average cosine similarity is relatively low (0.4-0.65), but within a few steps of training for all models, it sharply increases to $> 0.9$. These results remain similar if we use random sampling instead of greedy decoding, and other datasets like GSM8K and ARC-Challenge (Appendix E). Further, the outputs for many prompts in the greedy decoding case are trivial repetitions of the same token, e.g., newline '\n', a clear manifestation of repetition bias.

Similar representation collapse patterns as Pythia are observed for the OLMo-2 7B model. For its earliest available training checkpoint (step 150, `OLMo-2-1124-7B`), the average representation cosine similarity in the setup from Section 6 is $\approx 0.93$; for the next checkpoint at step 600, this value has already decreased to $\approx 0.43$ (similar for both greedy decoding and random sampling strategies). *Hence, repetition bias and representation collapse occur in the early pre-training phase of LLMs, validating our findings beyond toy settings.*

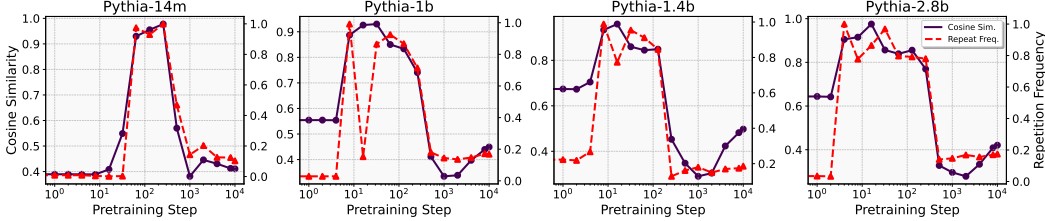

Figure 8: **Representation Collapse, Repetitions during Pythia Pretraining.** Representation collapse at different Pythia pretraining checkpoints evaluated on the ARC-Easy test dataset; tokens generated via greedy decoding. We find that similar to our toy experiments, average pairwise cosine similarity between hidden states and repetition frequency start at relatively low values, and increase to near 1.0 during the early phase of pretraining, followed by a decrease around step 1000.

# 7 Related Work

Abrupt learning has been studied in various settings; [9] study sudden drop in loss when training BERT, and show abrupt learning of the attention map (termed Syntactic Attention Structure / SAS) concurrent with the sudden drop in training loss. [20] study sudden drop in loss ('Eureka moments') when training Transformers on multi-step tasks. They show that ill-distributed softmax attention scores lead to small gradients for key, query weights causing slow learning of attention map, and that appropriately modifying softmax temperature can alleviate optimization issues. [3] studied abrupt learning in various neural net architectures trained on sparse parity tasks, demonstrating underlying hidden progress during training via measures such as Fourier gap and weight movement. [18] study abrupt learning when training a BERT model on matrix completion, while [32] analyze abrupt learning for a grammar data setup through graph percolation. For abrupt learning in in-context learning [17], there has been a line of recent works [10, 15, 41, 44, 47, 54, 55] that proposed various theoretical and empirical explanations. [15] studied abrupt learning for a Markov chain in-context learning task with Transformers, and demonstrate a partial solution in their setup in the form of a unigram estimator. [46] analyze partial solutions for an in-context $n$–gram task with Transformers, corresponding to $k$–gram estimators ($k < n$). For diagonal-attention Transformers with small initialization, [7] analyze 'saddle-to-saddle' dynamics for a theoretical understanding of loss plateaus during training.

A line of recent work has focused on understanding 'grokking' [39], i.e. abrupt generalization after an extended phase of memorization of training data by the model. Subsequent works have studied grokking through circuit formation [34, 36, 45], representation learning [31], delay in feature learning [27, 33, 52]. [40] further study 'naive loss minimization' and numerical stability issues for understanding grokking. Note that grokking is a fixed-dataset phenomenon concerning memorization of the training dataset, and hence somewhat distinct from abrupt learning studied in this paper.

Rank collapse is a related phenomenon for deep softmax transformers at initialization that might hinder training [2]; however, our representation collapse phenomenon is different in that (i) we use shallow (1 or 2 layers) Transformers instead of deep ones; (ii) we use linear attention instead of softmax; (iii) our observed representation collapse occurs only *after* a few steps of training, not at initialization. [4] show a form of representation collapse so that for 2 sequences $(v_1, v_2, \ldots, v_n)$ and $(v_1, v_2, \ldots, v_n, v_n)$, as $n$ grows large, the pretrained model's hidden state representation for the last token becomes identical for both sequences (Theorem 4.2, [4]). Note that we study evolution of representation collapse in the early phase of Transformer training, distinct from the notion of representation collapse at initialization, or in final pretrained models.

Repetition in language model outputs is a well studied problem [16, 19, 21, 23, 30, 48, 51, 53]. However most works focus not on the early phase of training, but on how repetition may arise in trained language models, and how to mitigate them. Towards understanding learning dynamics, [8, 11] report token repetitions in the early phase of training language models. Please see Appendix I for additional discussion on related work.

# 8 Discussion

We identified repetition bias and representation collapse as key characteristics of the early-phase implicit biases of Transformer training, which are closely connected to the commonly observed loss plateau. We further discussed 'search' over sequence tokens as a possible cause for loss plateau. The question of *why* behaviors such as representation collapse exist during early time training is an important question for future work. Furthermore, while we hypothesize a search-like phenomenon to be the reason behind slow learning of attention map, a rigorous theoretical validation of this hypothesis, and how it possibly connects to the intuitive "complexity" of the task, are important questions for further research.

**Acknowledgments**  This work was supported by DARPA. The authors thank Misha Belkin, Ekdeep Singh Lubana, Yongyi Yang, Zhiwei Xu, Vaibhav Balloli, and anonymous reviewers for helpful comments at various stages of this work. Part of this work was done when the authors were visiting the Modern Paradigms in Generalization Program at the Simons Institute for the Theory of Computing.

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

# A   Implementation Details

**Compute Resources.**   All experiments were conducted on a single GPU (NVIDIA A100 or L40S) on an academic computing cluster. Most training runs in this paper complete within a few hours.

**Causal Linear Attention.**   Linear attention transformer is obtained simply by removing the softmax activation function when computing the attention map, and setting the causal mask to $0$ instead of $-\infty$. We use the existing minGPT implementation [26] (MIT licence) for our experiments, modifying the code as above and wherever required.

**LLM Experiments.**   We use Pythia [6] / OLMo-2 [37] pretrained models (Apache 2.0 Licence) hosted on Huggingface Transformers [50] and evaluate them on the ARC-Easy / Challenge datasets [12] (CC-BY-SA 4.0 Licence), and GSM8K [13] (MIT Licence). We set the `use_cache=False` in the `generate` function, and use the hidden state used for predicting each of the 8 output tokens. For random sampling, we use `do_sample=True` (using default temperature value), using `do_sample=False` for our greedy decoding results.

# B   Results for Other Algorithmic Tasks

This section presents results on a suite of algorithmic tasks, verifying the generality of our identified phenomena.

Table 1: Algorithmic tasks that show abrupt learning and partial solution during plateau

| Task | Description | Partial Solution |
|---|---|---|
| Moving Window Sum (MWS) | Sum over moving window of 2 elements, copy 1st element | First input element |
| Prefix Sum (PRE) | Compute prefix sum of a given $n$–length sequence | First input element |
| Permutation (PER) | Permute an $n-$length sequence by given permutation | Incorrect permutation of input sequence |
| Multi-Digit Addition (ADD) | Add atmost–$n$–digit numbers | First digit (0 or 1) i.e. total carry-over from $n$ digits |
| Histogram (HIST) | Compute counts of each element in $n$–length sequence | $\approx 100\%$ Repetitive sequences |
| Reverse (REV) | Reverse $n$–length input sequence | Repetitive sequences[1] |
| Copy (COPY) | Copy $n$–length input sequence | Repetitive sequences[1] |
| Copy + MWS + MWS$_3$ | Concatenation of sequences from COPY, MWS, and MWS$_3$ (moving window of 3 elements) | During 1st plateau: COPY part correct; 2nd plateau: COPY and MWS parts correct (Appendix B.7) |

([1]*The loss plateau is very brief, hence a partial solution like other cases is not applicable.*)

In the following table we describe APM sets $\Omega$ for different algorithmic tasks used in this work. We assume the attention map is of the shape $A \in R^{(L-1)\times(L-1)}$ in the next token prediction setup on the full input sequence. Hence, for MWS, Prefix sum, Histogram, Reverse, Copy, $L = 33$, for permutation $L = 50$, and for Multi-digit addition $L = 15$. The APM sets are denoted by sets of tuples $(i, j) \in [L-1] \times [L-1]$ indicating the row/column indices in the attention map $A$.

Table 2: Attention Progress Measure (APM) sets ($\Omega$) for all tasks

| Task | APM set $\Omega$ |
|------|------------------|
| MWS | $\{(16,0)\} \cup \{(16+i, i-1) \mid i \in [1,15]\} \cup \{(16+i,i) \mid i \in [1,15]\}$ |
| Prefix Sum | $\{(16,0)\} \cup \{(16+i, 16+i) \mid i \in [1,15]\} \cup \{(16+i,i) \mid i \in [1,15]\}$ |
| Multi-digit Addition | $\{(i, 12-i) \mid i \in [9,12]\} \cup \{(i, 17-i) \mid i \in [9,12]\} \cup$ $\{(i, 13-i) \mid i \in [9,13]\} \cup \{(i, 18-i) \mid i \in [9,13]\}$ |
| Permutation | Layer 1: $\{(17+i, \pi_{i+1}-1) \mid i \in [0,15]\}$ |
| | Layer 2: $\{(33+i, 17+i) \mid i \in [0,15]\}$ |
| Histogram | Layer 1: $\{(16+i, i) \mid i \in [0,15]\}$ |
| Copy | $\{(16+i, i) \mid i \in [0,15]\}$ |
| Reverse | $\{(16+i, 15-i) \mid i \in [0,15]\}$ |

## B.1 Multi-Digit Addition

This task involves adding 2 atmost 4–digit numbers; if the numbers are represented as $a = \overline{a_1 a_2 a_3 a_4}$, $b = \overline{b_1 b_2 b_3 b_4}$ and their sum $a + b = c = \overline{c_0 c_1 c_2 c_3 c_4}$ then the training sequences for ADD are of the form

$$a_1, a_2, a_3, a_4, +, b_1, b_2, b_3, b_4, =, c_4, c_3, c_2, c_1, c_0$$

Note that the output sequence is reversed, following the observations from [28]. We find similar abrupt learning characteristics (Figure 9), partial solution in this case being $c_0$ i.e. total carry-over from 4 single digit add operations. An interpretable attention map learnt for the output sequence is shown in Figure 10.

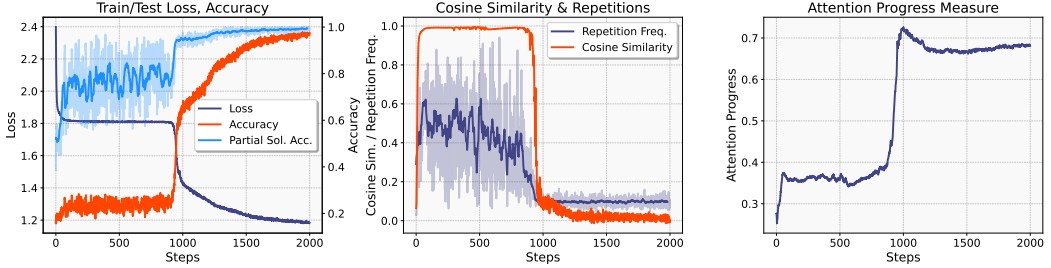

Figure 9: Training dynamics for Add task. (left) Train/Test Loss, Accuracy and Partial solution progress ($c_0$ accuracy); (middle) Repetition frequency and representation collapse; (right) Attention progress measure.

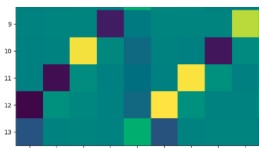

Figure 10: Attention map for add task, note that the model attends to the relevant digits in the input numbers, and to somewhat lesser extent to the preceding digits as well (highlighted positions show entries with larger magnitude).

## B.2 Prefix sum

This task involves computing the cumulative (prefix) sum of an $n-$length sequence of integers, so that the training sequences in PRE are of the form ($n = 16, \text{SEP} = 17$),

$$x_1, x_2, \ldots, x_n, \text{SEP}, y_1, y_2, \ldots, y_n$$

$$y_i = \left(\sum_{j=1}^{i} x_j\right) \bmod 17 \quad \forall i \in [n]$$

Training dynamics for this task are shown in Fig. 11 which show similar abrupt learning behavior as MWS and partial solution learning for $y_1$. The interpretable attention map learnt for this task is shown in Figure 12.

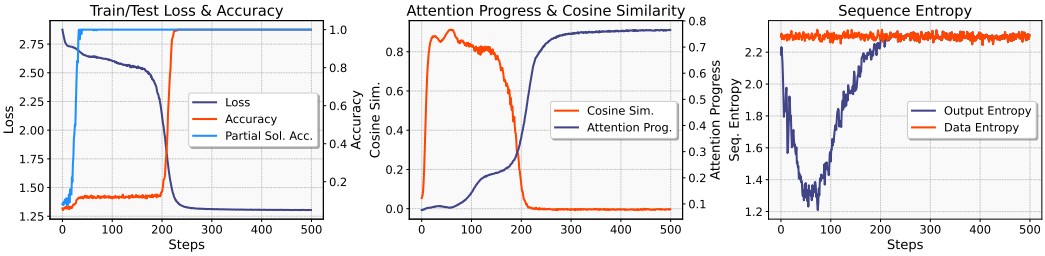

Figure 11: Training dynamics for Prefix sum task. (left) Train/Test Loss, Accuracy and Partial solution progress ($y_1$ accuracy); (middle) Attention progress and representation collapse; (right) SeqEnt for data and model output sequences.

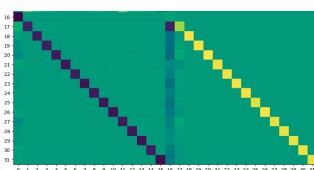

Figure 12: Attention map for Prefix sum task, that uses the relevant token in the input, as well as the previous token in the output to track prefix sum (highlighted positions show entries with larger magnitude).

## B.3 Permutation

This task involves training a 2-layer, 1-head Transformer on permuting a length$-n$ sequence using the permutation $\pi$, which is generated at random and is distinct for each training sequence. Formally, for a sequence of positive integers $(x_1, \ldots, x_n)$ and a permutation $(\pi_1, \ldots, \pi_n)$ over $[n]$, training sequences for PER, $k = 0, 1, 2, \ldots$ are given by

$$x_1, \ldots, x_n, \text{SEP}, \pi_1, \ldots, \pi_n, \text{SEP}, x_{\pi_1}, \ldots, x_{\pi_n}$$

where $x_i \sim \text{Unif}\{17, 18, \ldots, 32\}, n = 16, \text{SEP} = 0$. The partial solution in this case is the output sequence being an permutation of the input sequence $x_1, \ldots, x_n$ i.e., it learns to copy the tokens correctly, but in wrong order (Figure 13). The interpretable attention maps in this case show that the model learns to copy the correct tokens based on the permutation provided (Figure 14a) and then uses them for the final output sequence (Figure 14b).

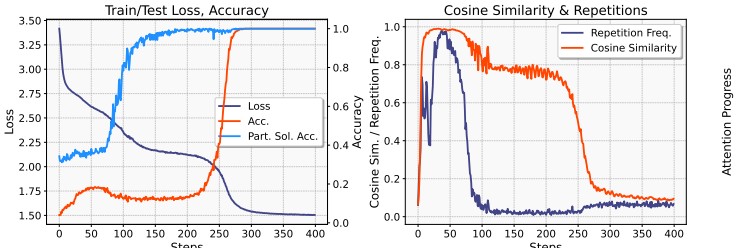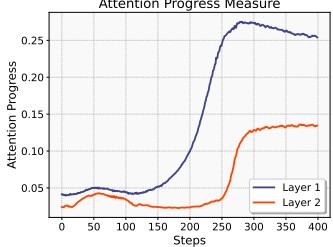

Figure 13: Training dynamics for Permutation task. (left) Train/Test Loss, Accuracy and Partial solution progress; (middle) Repetition frequency and representation collapse; (right) Attention progress measure. Note that the repetition frequency decreases by step 100, which is followed by the partial solution.

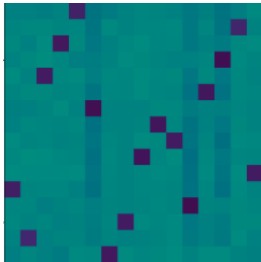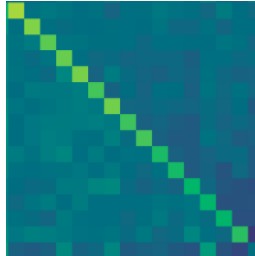

(a) Attention map in Layer 1 where rows are attention weights over the input part $x_1, x_2, \ldots, x_n$ of the sequence. The highlighted positions are attending to $\pi_1, \pi_2, \ldots, \pi_n = 5, 15, 4, 14, 3, 13, 6, 10, 11, 9, 16, 1, 12, 8, 2, 7$. for index $i \in [n]$.

(b) Attention map in Layer 2; the rows (output part of the sequence) are attention scores over the part of sequence to which Layer 1 attention map copies the correctly permuted tokens. This implies that this attention map simply copies the correct token from the residual stream after Layer 1.

Figure 14: Attention maps for the 2 layer Transformer used for Permutation task; highlighted positions show entries with larger magnitude.

## B.4 Histogram

This task [14] involves computing the counts of elements in the input sequence, and training sequences are of the form

$$x_1, x_2, \ldots, x_n, \text{SEP}, y_1, y_2, \ldots, y_n$$

$$y_i = \sum_{j=1}^{n} \mathbf{1}[x_j = x_i]$$

where $x_i \sim \text{Unif}\{1, 2, \ldots, 12\}, n = 16, \text{SEP} = 0$. We train a 2-layer, 1-head transformer for this task, with gradient clipping $(1.0)$ to avoid loss spikes (Figure 15). We note that the repetition bias in this case is quite strong which leads to $\approx 100\%$ repetitions in the early phase of training, and which we characterize as partial solution for this task. Further we only consider the attention map from layer 1 (Figure 16) since this is the most consistent and clearly interpretable across runs, and indicates an identity-map-like function.

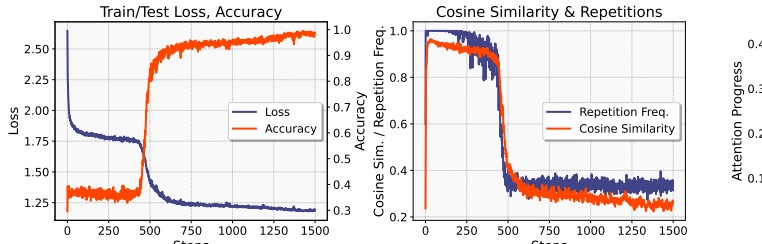 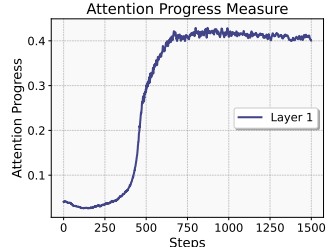

Figure 15: Training dynamics for Histogram task. (left) Train/Test Loss and Accuracy; (middle) Repetition frequency and representation collapse; (right) Attention progress measure. We only measure attention progress for the 1st layer, since that is the one that consistently and clearly shows an interpretable pattern (Figure 16).

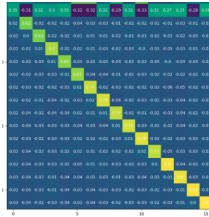

Figure 16: Attention map in layer 1 for histogram task, where rows for the latter half of the sequence compute attention weights over the input tokens $x_i$, similar to an identity map (highlighted positions show entries with larger magnitude).

## B.5 Reverse

This is the task of reversing the input sequence, so that the training sequences for reverse task REV are given as,

$$x_1, x_2, \ldots, x_n, \mathrm{SEP}, x_n, x_{n-1}, \ldots, x_1$$

for $x_i \sim \mathrm{Unif}\{1, 2, \ldots, 16\}, n = 16, \mathrm{SEP} = 0$. The training dynamics are shown in Figure 17a and the interpretable attention map is shown in Figure 17b.

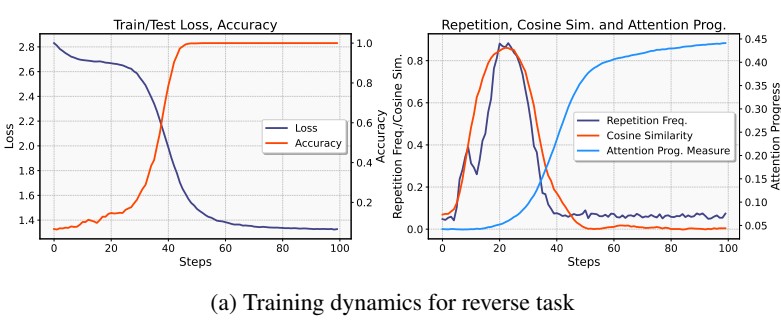 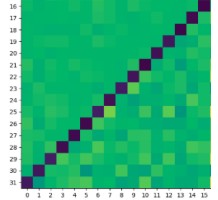

(b) Attention map for reverse task (highlighted positions show entries with larger magnitude).

(a) Training dynamics for reverse task

Figure 17: **Training dynamics for Reverse task.** We see Abrupt Learning, Representation Collapse and Repetitions, though to a lesser extent than MWS task. Note that the plateau is much shorter compared to MWS, possibly explained by the fact that reversing a sequence is 'easier' than computing the moving window sum.

## B.6 Copy

This is the trivial task of copying the input sequence as is, so that the training sequences for copy task COPY are given as,

$$x_1, x_2, \ldots, x_n, \mathrm{SEP}, x_1, x_2, \ldots, x_n$$

for $x_i \sim \mathrm{Unif}\{1, 2, \ldots, 16\}, n = 16, \mathrm{SEP} = 0$. The training dynamics are shown in Figure 18a and the interpretable attention map is shown in Figure 18b.

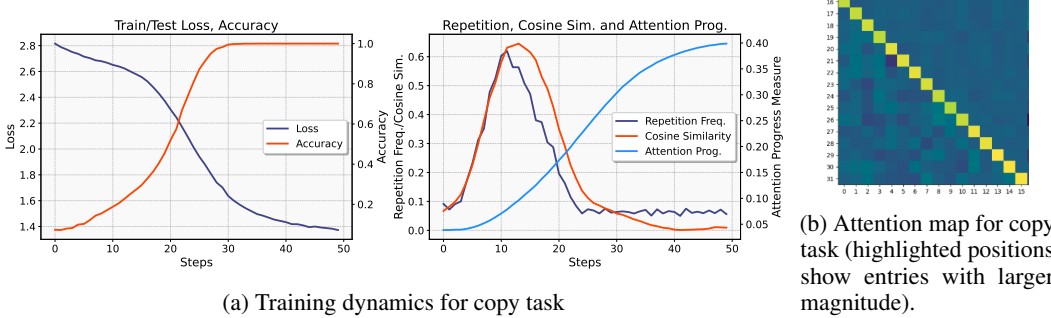

(a) Training dynamics for copy task

(b) Attention map for copy task (highlighted positions show entries with larger magnitude).

Figure 18: **Training dynamics for Copy task.** Similar to reverse task, we observe Abrupt Learning, Representation Collapse and Repetitions for Copy task, but this time to an even lesser extent than reverse task itself.

### B.7 Concatenation of Copy, MWS, MWS$_3$

We train the model on following sequences,

$$x_1, x_2, \ldots, x_{16}, \mathrm{SEP}, y_1, y_2, \ldots, y_{13}$$

$$y_i = \begin{cases} x_i & i \in [1, 4] \\ (x_i + x_{i+1}) \bmod 17 & i \in [5, 9] \\ (x_{i+1} + x_{i+2} + x_{i+3}) \bmod 17 & i \in [10, 13] \end{cases}$$

with $x_i \sim \mathrm{Unif}\{1, 2, \ldots, 16\}, \mathrm{SEP} = 17$, and find that the train loss follows a 'staircase'-like evolution, learning the 3 distinct parts $([y_1, \ldots, y_4], [y_5, \ldots, y_9], [y_{10}, \ldots, y_{13}])$ sequentially (Figure 19). Similar sequential trend is seen for accuracy and average cosine similarity, measured separately over these parts (labeled Part 1,2,3).

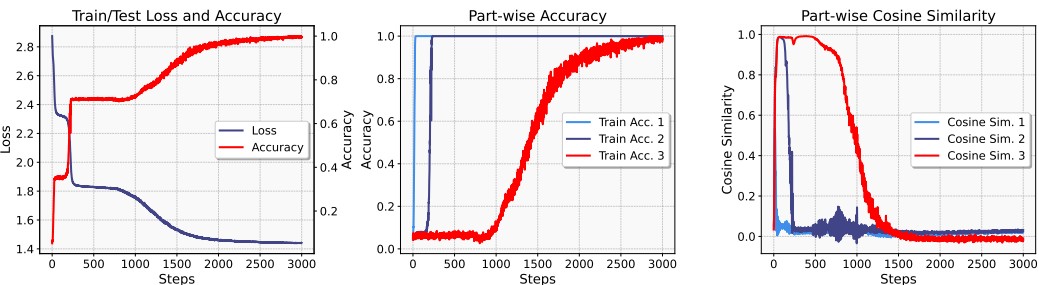

Figure 19: **Partial Solution for Concatenated Target Sequence.**

## C  Probing Attention Head Output

We probe the output hidden states of the Attention block, before they are added to the residual stream (Equation (2)). Specifically, we use a 2-layer MLP probe with hidden layer width $256$ and ReLU activation function to map these hidden states to the ground truth output labels $y_i$. Formally, for hidden states $\mathbf{h} \in \mathbb{R}^{256}$, the probe is defined as $\mathrm{Probe}(\mathbf{h}) := W_{P,2}\mathrm{ReLU}(W_{P,1}\mathbf{h})$, $W_{P,2} \in \mathbb{R}^{17 \times 256}, W_{P,1} \in \mathbb{R}^{256 \times 256}$, implemented using scikit-learn `MLPClassifier` [35]. We probe at every 10 training steps of the Transformer model, training the probe on $1024$ samples and evaluating using $1024$ held-out samples, both different from training data for the Transformer.

*We find that the probe test accuracy increases earlier than the sudden jump in model accuracy (Figure 20), demonstrating 'hidden' progress in Attention block outputs.*

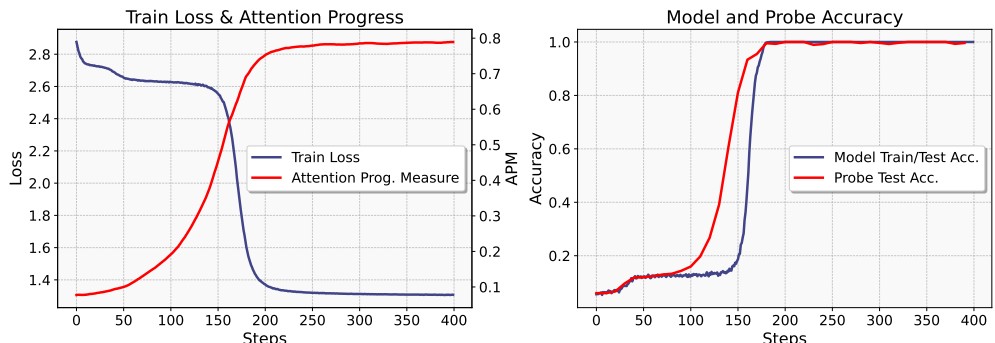

Figure 20: **Probing Attention Outputs.** We find that the probe accuracy when probing attention block outputs (Equation (2)) starts to increase earlier than the model accruacy during training, highlighting 'hidden' gradual progress in the hidden states in addition to attention map (via attention progress measure).

# D  Results for REPEAT$_2$, REPEAT$_4$

**REPEAT$_2$**    This task is defined as,

$$y_i = \begin{cases} x_1 & 1 \leq i \leq 8 \\ (x_1 + 1) \bmod 17 & 9 \leq i \leq 16 \end{cases}$$

The training dynamics for REPEAT$_2$ are given in Figure 21a. We find that similar to REPEAT$_1$, the training loss does not exhibit any plateau. Moreover, the repetition frequency $\rho$ and metric $\alpha_1$ increase rapidly to $\approx 1.0$ early on in training. We measure the average cosine similarity for hidden states for repetitive blocks of the output sequence (see Figure 21b).

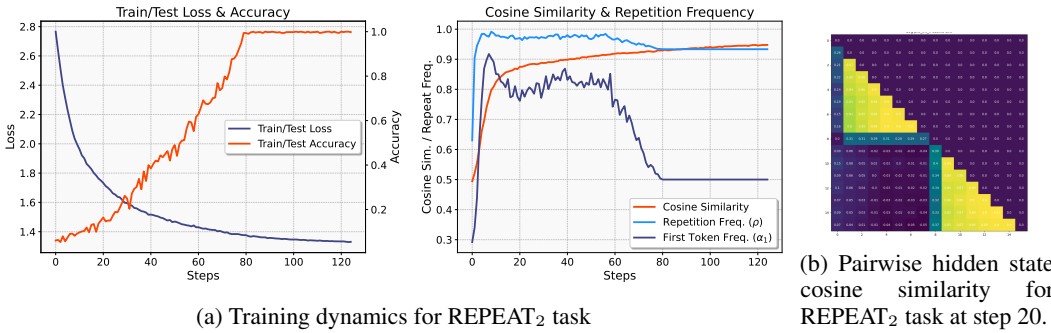

(a) Training dynamics for REPEAT$_2$ task

(b) Pairwise hidden state cosine similarity for REPEAT$_2$ task at step 20.

Figure 21: **REPEAT$_2$ training dynamics.** Note that there is no plateau in loss, similar to the REPEAT$_1$ task. Further, the pairwise cosine similarity for hidden states take a specific form indicating the blocks of repeated tokens in the output.

**REPEAT$_4$**    This task is defined as

$$y_i = \begin{cases} x_1 & 1 \leq i \leq 4 \\ (x_1 + 1) \bmod 17 & 5 \leq i \leq 8 \\ (x_1 + 2) \bmod 17 & 9 \leq i \leq 12 \\ (x_1 + 3) \bmod 17 & 13 \leq i \leq 16 \end{cases}$$

Similar results are observed for REPEAT$_4$ as well (Figure 22a); for this case we measure the average cosine similarity for hidden states for repetitive blocks of the output sequence (see Figure 22b).

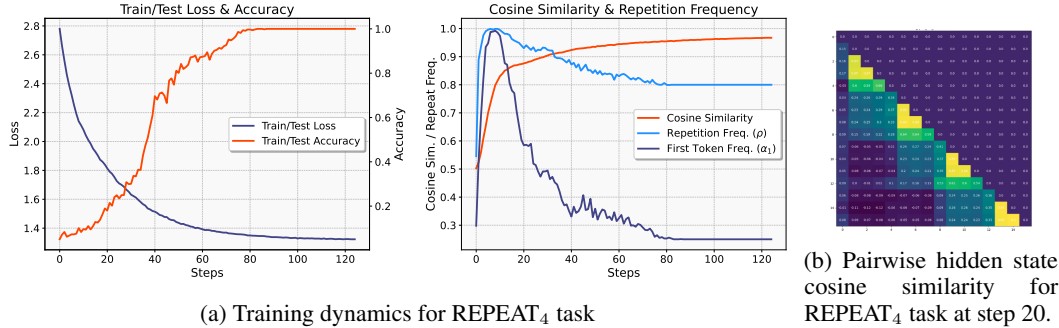

(a) Training dynamics for REPEAT$_4$ task

(b) Pairwise hidden state cosine similarity for REPEAT$_4$ task at step 20.

Figure 22: **REPEAT$_4$ training dynamics.** Similar to REPEAT$_2$, there is no plateau in loss. The pairwise cosine similarity for hidden states takes a form indicating the blocks of repeated tokens in the output.

# E    Additional LLM Results

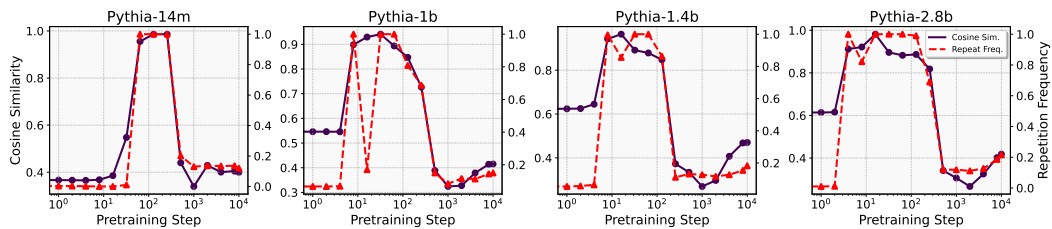

Figure 23: **Representation Collapse, Repetitions during Pythia Pretraining (GSM-8K [13]).** Representation collapse at different Pythia pretraining checkpoints evaluated on the GSM-8K test dataset; tokens generated via greedy decoding.

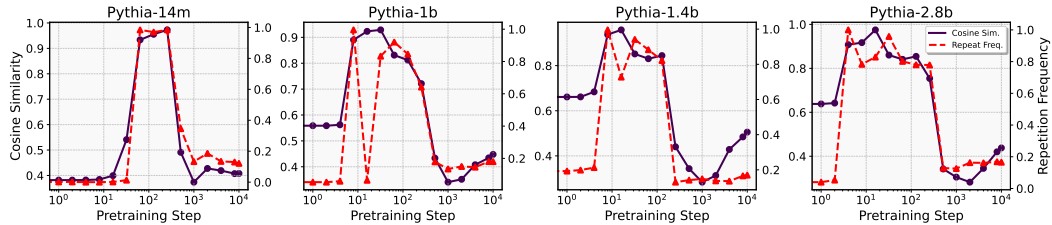

Figure 24: **Representation Collapse, Repetitions during Pythia Pretraining (ARC-Challenge [12]).** Representation collapse at different Pythia pretraining checkpoints evaluated on the ARC-Challenge test dataset; tokens generated via greedy decoding.

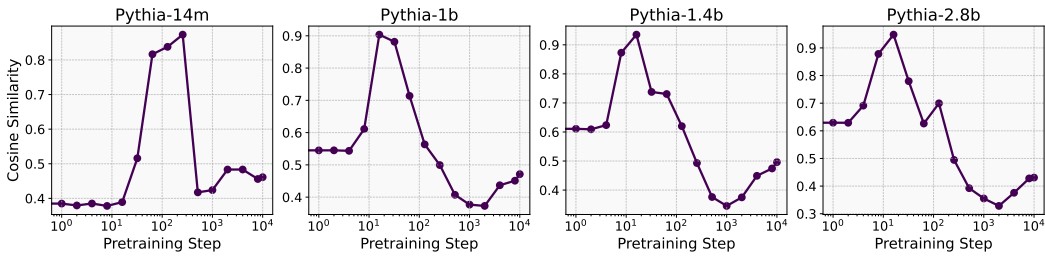

Figure 25: Representation collapse during Pythia pretraining (inference with random sampling).

# F Varying Model Configurations

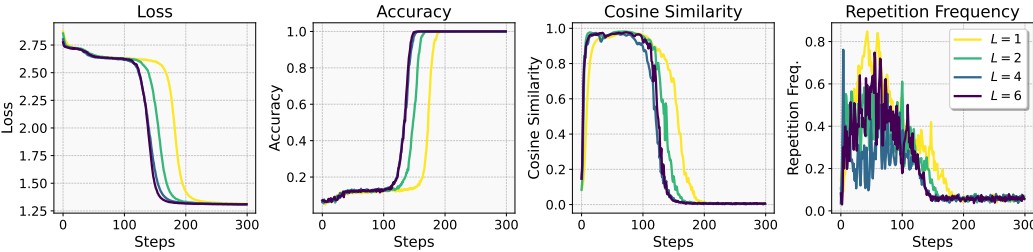

Figure 26: **Number of Layers** ($L$). Abrupt learning, representation collapse in the last layer and repetitions for multi–layer models.

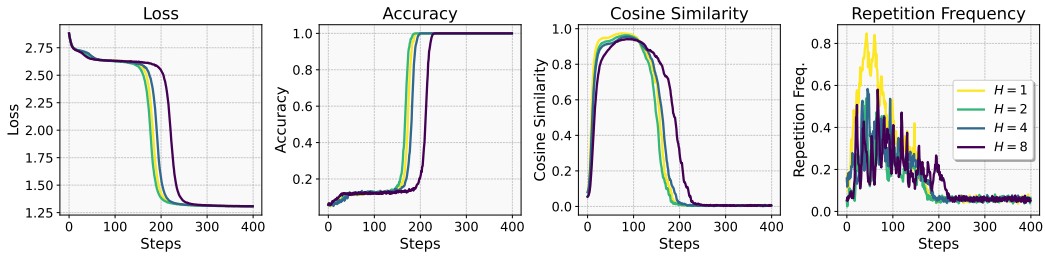

Figure 27: **Number of attention heads** ($H$). Representation collapse and repetition bias in 1-layer multi-attention head models (embedding dimension is fixed at $d = 256$ for all cases).

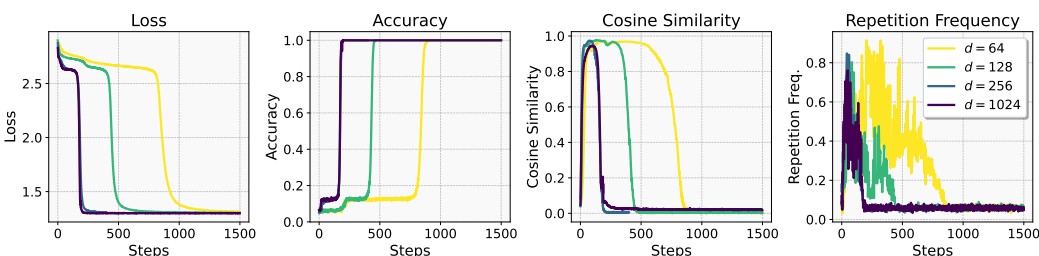

Figure 28: **Embedding dimension** ($d$). Abrupt learning with representation collapse and repetition bias in 1-layer 1-head models with different embedding dimensions.

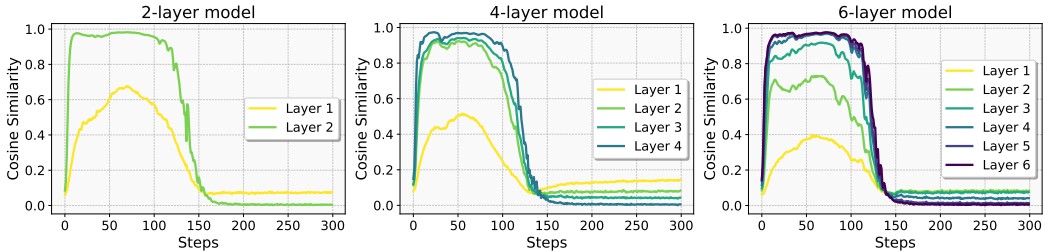

Figure 29: **Extent of Representation collapse at various intermediate layers.** Cosine similarity values showing the extent of representation collapse after each intermediate layer in multi-layer models. Note that the representation collapse is not so severe in the early layers of multi-layer models, but the cosine similarity becomes close to $1.0$ as we progress to the final layer.

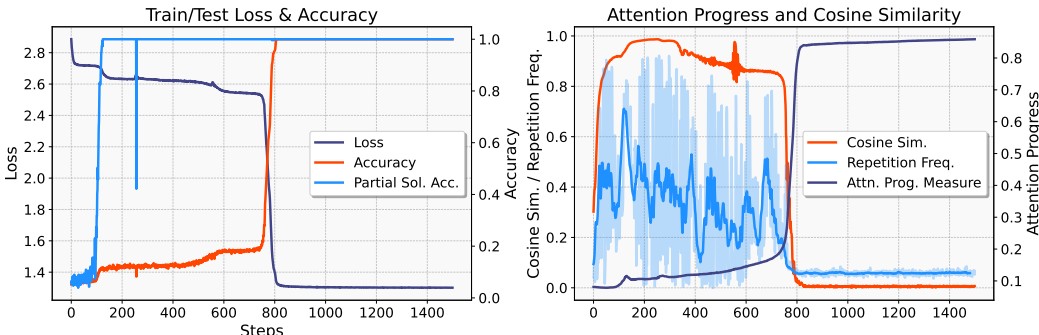

Figure 30: **Softmax Attention**. For completeness we show that repetition bias and early-phase representation collapse are not limited to linear transformers but are observed in softmax attention transformers as well. Note that the loss plateau is longer than that for linear attention.

# G   Varying Optimization Setups

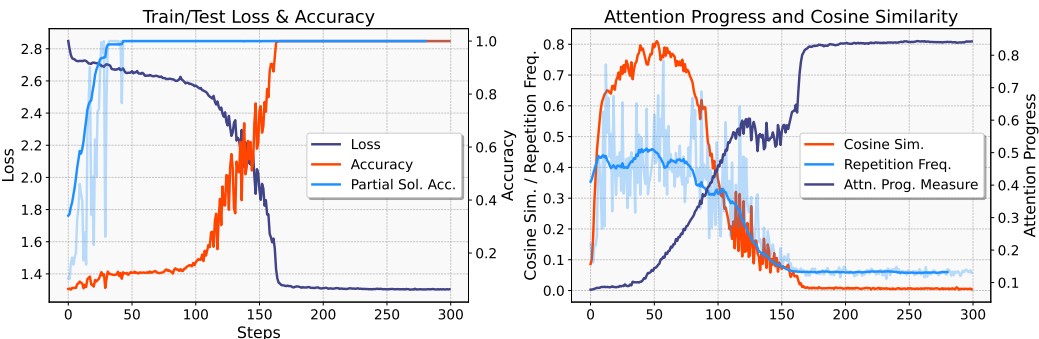

Figure 31: **Using SGD for training**. We show that abrupt learning is not limited to Adam optimizer, and occurs with SGD ($\eta = 0.1$) as well. We chose this value of $\eta$ since smaller values typically lead to much longer periods of little decrease in loss, without increase in accuracy.

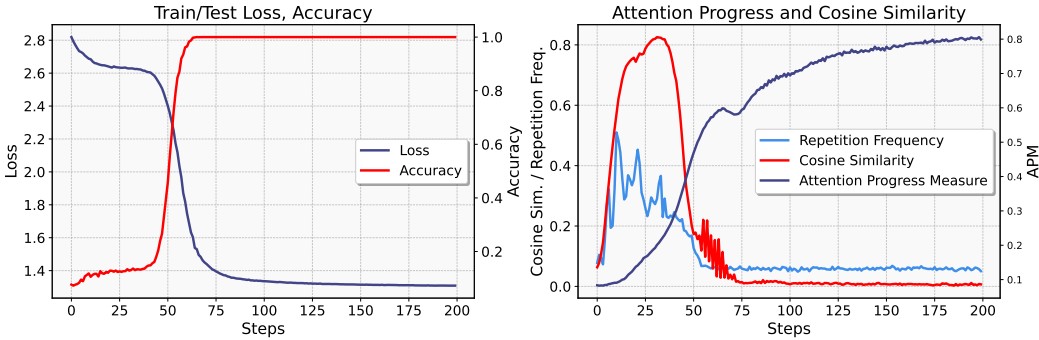

Figure 32: **Using Muon for training, with Adam for embeddings, LM head and bias parameters**. We use learning rate $0.02$ for Muon, and 1e–4 for Adam. Note that while the plateau duration decreases significantly compared to using only Adam (Figure 2a), a plateau of duration $\sim 40$ steps still occurs.

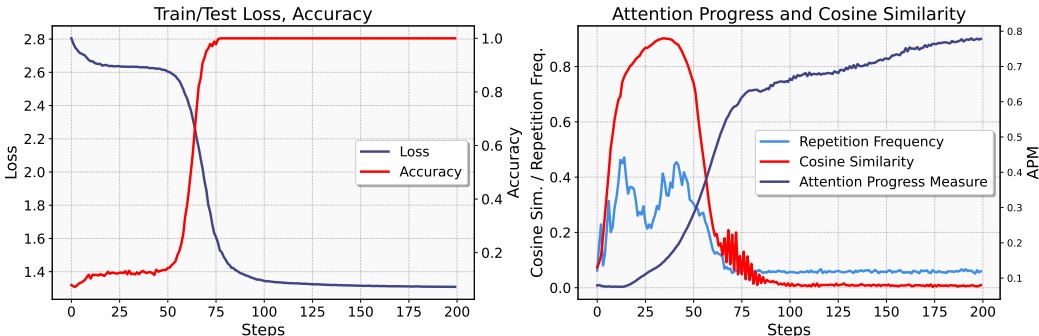

Figure 33: **Using Muon for training, with AdamW for embeddings, LM head and bias parameters**. We use learning rate $0.02$ for Muon, and 1e–4 for AdamW (weight decay$= 0.01$).

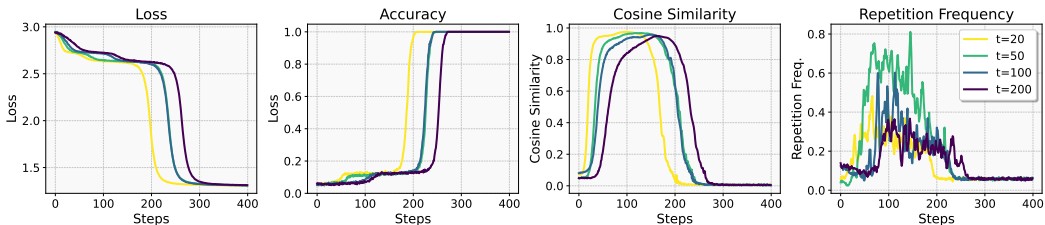

Figure 34: **Training Dynamics with Learning Rate Warmup.** The learning rate increases linearly from 0 to 1e–4 during training steps $[0, t]$.

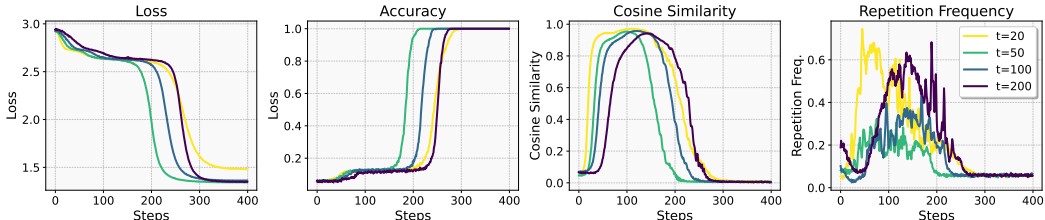

Figure 35: **Training Dynamics with Learning Rate Warmup + Cosine Decay.** The learning rate increases linearly from 0 to 1e–4 during training steps $[0, t]$, and decreases as a cosine function during steps $[t + 1, 400]$.

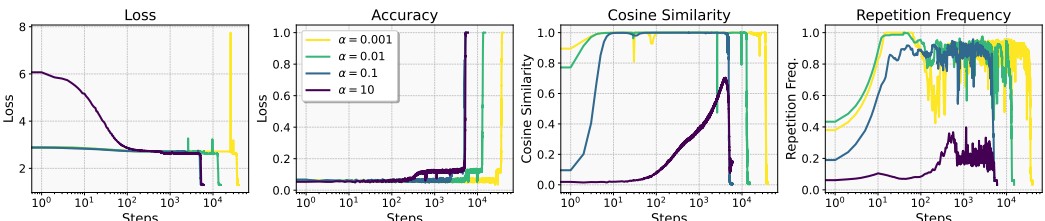

Figure 36: **Training Dynamics with different Initialization Scales.** We find that smaller initialization scale ($\alpha < 1$) leads to a longer plateau, and cosine similarity being very close to $1.0$ throughout the plateau. With $\alpha = 10$, the representation collapse is not as strong, reaching a peak value of $\approx 0.6$. We smoothen the repetition frequency curve with window of size 20 for presentation clarity, using `np.convolve(repeat_freq, np.ones(20)/20, mode='same')`.

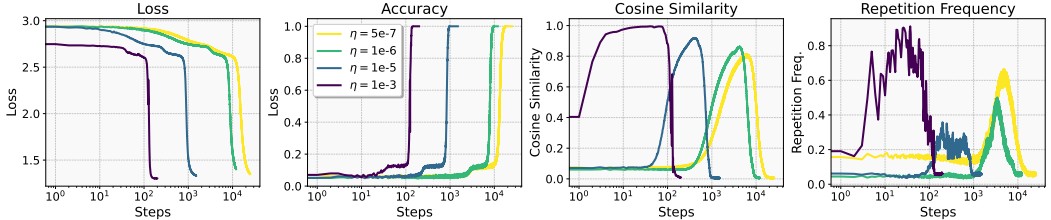

Figure 37: **Training Dynamics with different Learning Rates.**

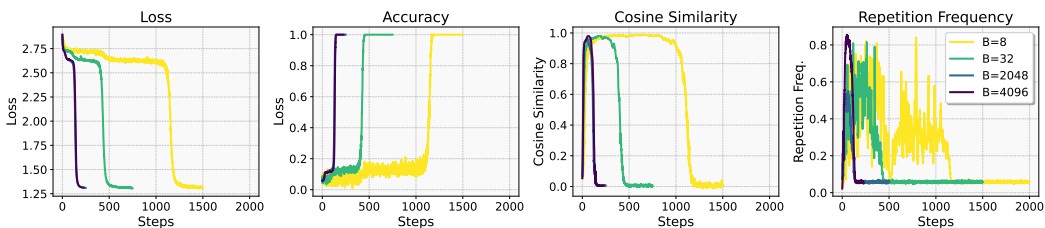

Figure 38: **Training Dynamics with varying Batch Size.**

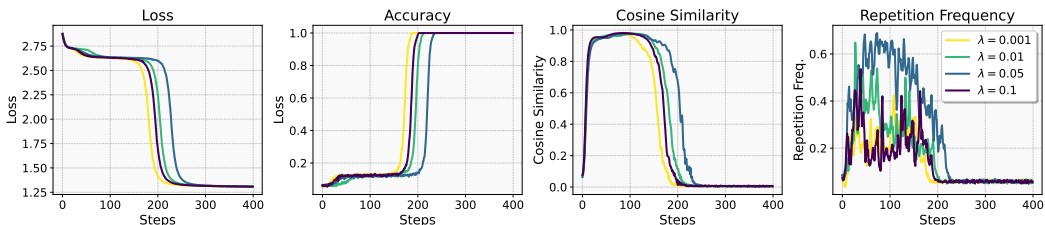

Figure 39: **Training Dynamics with AdamW, varying weight decay parameter** $(\lambda)$.

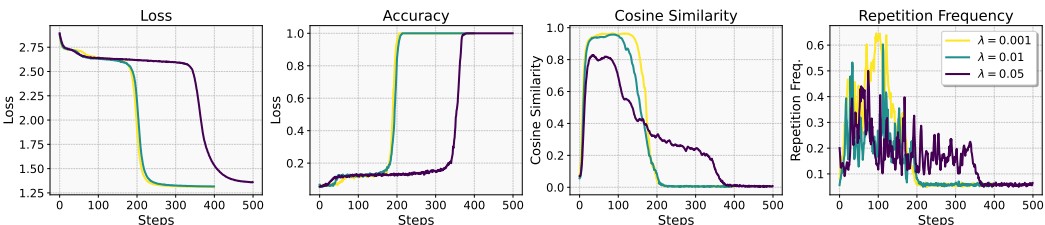

Figure 40: **Training Dynamics with $\ell_2$ regularization, varying regularization parameter** $(\lambda)$.

## H  Additional Figures

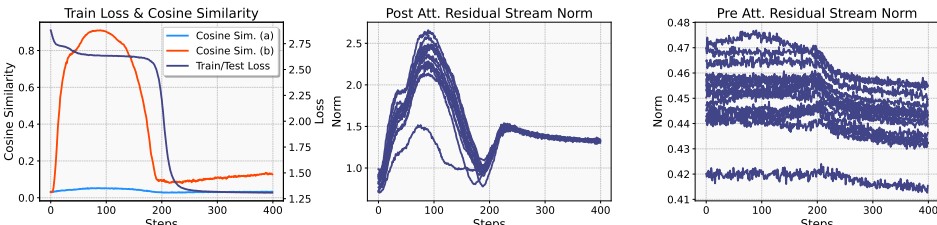

Figure 41: Norm and representation collapse dynamics for (a) pre- and (b) post-attention residual streams for all positions $i, j$ except the first position.

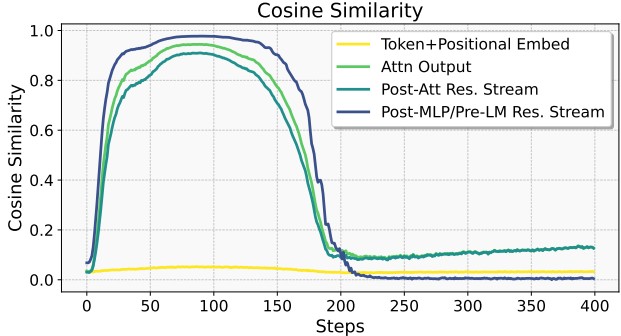

Figure 42: Cosine similarity at various points in residual stream for 1-layer, 1-head Transformer trained on MWS task.

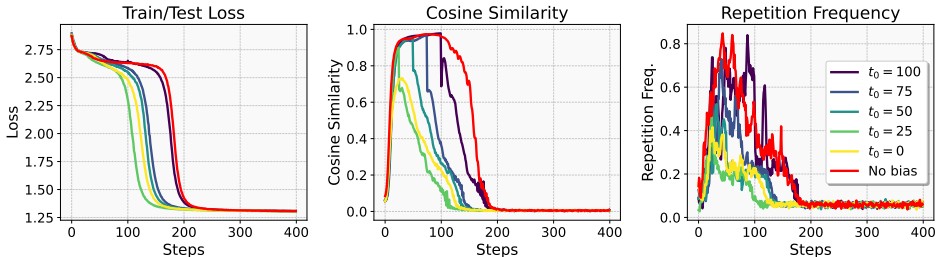

Figure 43: Biasing attention map by $c = 10$ at different $t_0$ during training.

## I  Additional Related Work

Techniques from statistical physics have also been used towards understanding initial loss plateaus in neural net training [24, 43]; they work in a 2-layer teacher-student setup, where the second layer is fixed during training, and use order parameters to study training. They show that there is a permutation symmetry in the weight vectors of the first layer during the early plateau stage, and exiting this state leads to drop in loss. Singular learning theory [22] has also been used to explain stagewise learning dynamics in Transformers; they estimate the 'Local Learning Coefficient' (LLC) during training to quantify degeneracy in the loss landscape, and consequently explain the learning dynamics. The interplay of simplicity bias and Transformer learning dynamics has also been studied recently in [5, 42]. [42] show that Transformers progressively learn higher-order ('many-body') interactions among tokens in the sequence. In [5] authors show that neural nets learn to use lower-order moments of data early in training via interventions on test data, and show an equivalence between $n-$gram statistics and embedding moments for discrete domains.

