# OpenReview forum: "What Happens During the Loss Plateau? Understanding Abrupt Learning in Transformers"
_NeurIPS.cc/2025/Conference — NeurIPS 2025 poster_

### Official Review · Reviewer_dDr4 · 2025-06-17

**Clarity:** 4
**Significance:** 1
**Originality:** 2
**Rating:** 4
**Confidence:** 3

**Summary:**

In this paper, the authors investigate the phenomenon of abrupt learning (i.e., the sudden learning of algorithmic tasks such as moving-window-sum, prefix sum, etc.) in shallow Transformer models. In this setting, they empirically observe different phenomena during the loss plateau preceding the task solution: collapse of the internal representations, output repetition bias, and convergence to partially correct solutions. Additionally, the authors show that some of the identified behaviors (e.g., repetition bias and representation collapse) can also be observed in the early stages of the pre-training of larger models, such as Pythia and OLMo.

**Questions:**

1. Why did you not consider the sequence entropy as a measure of repetition bias from the beginning? If the number of tokens which are equal to the followings is high, I would also expect the entropy-based measure to capture this information. What is the point in introducing a flawed measure to then just change it afterwards?
2. How do you bias the attention matrix? Lines 183-186 are not clear.
3. It would be good to avoid color overlap between Figures 3 and 4 (which could be probably merged into a single figure as well?).
4. How do you tokenize the inputs? This is an important detail missing from the paper.

**Ethical Concerns:**

["NO or VERY MINOR ethics concerns only"]

**Final Justification:**

The authors clarified some of my initial concerns with additional experimental results and clarifications. However, still cannot recommend a full acceptance of the paper, as I still believe that the paper could have been made stronger by including more insights on why the observed phenomena emerge, to what extent they represent a real problem for language models, and how to alleviate them in practice. Also, I do not know to which extent the authors will address my concerns regarding issues in the manuscript itself; I sincerely hope that, in case of acceptance, all of the additional rebuttal results/discussion will be included in the paper.

**Limitations:**

Yes.

**Paper Formatting Concerns:**

No major formatting issues detected.

**Quality:**

2

**Strengths And Weaknesses:**

**Strengths**

The paper is overall well-written, and the narration is easy to follow. The authors evaluate a sufficiently wide choice of the algorithmic tasks (even if results for only one of them are included in the main text, see Weaknesses). The tasks and the [Transformer] models are very simple, allowing to make more tailored and precise observations on the nature of the observations. Furthermore, I also like that, at this level of complexity, it is possible to easily interpret some components of the model (such as the attention matrix) and intervene on them to validate hypotheses and conjectures (for instance, in the experiments on programming the correct attention mechanism vs. programming the optimal MLP/embeddings).


**Weaknesses**

1. The main weakness that I can identify in this paper is the relevance of the findings and the absence of a deeper investigation and understanding of them. For instance: I like the analysis on the internal representation collapse for MWS. However, why is this phenomenon emerging? Can this be tackled in a way which is not directly programming the “true” attention matrix for the model? The same goes for the repetition bias: I like how you somewhat exclude that this phenomenon purely emerges because of correlations in the data, but what is causing it then? Also, some of the results are unsurprising: the role of the attention mechanism in the representation collapse, for instance, since the attention weights at initialization will be uniformly distributed, resulting in an equally uniformly distributed mixing of the input tokens.
2. More in general, I do not see how the observations in the paper could contribute to a better understanding of grokking-like phenomena. While the authors claim in the related work “to aim to understand a unifying reason behind such observations […] without restrictive assumptions on our model or training setup”, they only provide a set of (valid) empirical observations and, in fact, use a very limited setup, which could be interpreted as using strong assumptions. Furthermore, in Line 292 they claim to also focus on “how to mitigate such phenomena” (repetition in language), which is not the case in this paper.
3. Claiming that the studied models converge to “interpretable partial solutions” in the early stages of the training is a bit stretched in my opinion. The claim relies on a single observation in MWS, where the model initially learns to copy the first token, which is probably not enough to generalize the statement so much. To claim this, the authors should include evidence at least all the other algorithmic tasks considered in the paper.
4. The extrapolation of the results to LLMs is also handwavy. The statements on repetition bias are not supported by any quantitative evidence. Also, the experimental setup (e.g., from which layer are the hidden states extracted, how are the 100 questions from ARC-Easy selected, etc.) for the representation collapse are not sufficiently described in the manuscript.
5. The results are not self-contained in the main paper. This is not a major weakness, but I believe that spending some time to fit more complete results directly in the main paper (and not in the appendix) could improve the manuscript.
6. It is not mandatory, but the authors could have included the experimental code in the submission. Given the simple nature of the setup, it would have been easy for me to replicate some of the experiments, which would have in turn increased my confidence in the reported results.

---

> ### Author Rebuttal · Authors · 2025-07-30
>
> We thank the reviewer for their positive comments, and are glad to know they found our paper well written and easy to follow. We address their concerns below.
>
> *Weaknesses*:
>
> 1. We would like to emphasize that our work contributes towards a unified understanding of abrupt learning in various algorithmic tasks for Transformers, through novel results about representation collapse in hidden states and repetitions in the output. We further show how learning attention map is a learning bottleneck, and causally verify this via training time interventions. Importantly, we also show how our findings about representation collapse and repetitions are applicable to LLM pretraining, highlighting the relevance of our work. Finally, these empirical results form the basis for future theoretical investigation into such phenomena in Transformer training.
>
> a) “*Why is this phenomenon emerging…*”: As we also note in the concluding discussion in the paper, this is an important direction for future work. We present some hypotheses for why these phenomena are occurring:
>
> **Hypothesis for slow learning of attention map**:
>
> Our hypothesis for loss plateau is that the attention layer first searches and combines the correct input tokens, and subsequently during training the MLP layer and LM head learn to map those hidden states to the correct output tokens. This is verified via non-linear (2-layer ReLU MLP) probing the outputs of Attention block during various points in training, and we find that such a probe is able to map the hidden states to the correct output labels earlier in training than the final model output (more details below). We note that this is related to prior work on the “search-then-fit” mechanism in 2–layer neural nets [Abbe 23], where the first layer in the neural network searches for the support variables of the target function in the input vector, and subsequently the 2nd layer fits the target function using these support variables.
>
> *Probing details*: We probe hidden state outputs $h_i$ of the Attention block corresponding to the output tokens, before they are added to the residual stream. We map these hidden states to the correct output tokens (i.e. $y_i$ as defined for MWS) using a 2-layer ReLU MLP probe ${R}^{256} \ni x  \mapsto W_2 \sigma(W_1 x) \in {R}^{17}, W_1 \in R^{256 \times 256}, W_2 \in R^{17 \times 256}$ mapping to logits over relevant vocabulary {0,...,16} (no SEP=17 token since that is not relevant for $y_i$). We implement the probe using MLPClassifier from the sklearn.neural_network library, train it using 1024 samples, and evaluate its performance using a held-out test set of 1024 samples.
>
> We will add this experiment to the updated version of the paper; a deeper theoretical understanding of this phenomenon is an important direction for future research.
>
>
> **Hypothesis for representation collapse**: We hypothesize that representation collapse arises from a bias of Transformers towards repetitive outputs in autoregressive setup; towards this hypothesis, we showed that training on repetitive sequences (hence explicitly encouraging representation collapse / repetition bias) is easier in the sense of training loss decreasing rapidly and monotonically, without any loss plateaus (Sec 5.2).
>
>
> b) “*Can this be tackled…*”: We have found that abrupt learning, representation collapse and repetition bias occur robustly across various tasks, hyperparameter choices and optimization settings, and how to tackle these phenomena without directly intervening on the attention map is an important question for future work.
>
> c) "*…results are unsurprising… uniformly distributed mixing of input tokens.*”: As we have also pointed out in Line 145 and 156, repetition bias / representation collapse phenomena are not present at initialization, and hence are not due to uniform attention weights at initialization. Instead, it emerges in the initial ~50 steps of training, which indicates it is a consequence of early phase model training. To quantitatively verify this, note that in Fig 2(b) representation collapse (cosine similarity) increases from 0.1 to 0.9, while repetition frequency also increases from around 0.1 to 0.8 in the first 50 steps of training.
>
> 2. “*Very limited setup … strong assumptions …*”: We note that our paper includes results on multiple tasks and model configurations (number layers, number of heads, etc.) for standard decoder-only Transformer architectures used in practice, confirming a unifying underlying behavior in all these cases. We also use a realistic next-token-prediction training setup with cross entropy loss and Adam optimizer, as is used in practice. In response to reviewer U6ru, we have also performed multiple ablations on the optimization setup and found abrupt learning, representation collapse and repetitions in all cases, highlighting that the phenomena are not limited to a single task / model / training setup. Our results on representation collapse and repetitions in LLM pretraining also demonstrate the generality of our findings in the toy model / data setup.
>
>     “*In Line 292 they claim*…": We would like to point out that we have not claimed to focus on mitigating repetition in LLM in our paper, and line 292 is part of the ‘Related Work’ section discussing other prior work on repetitions in LLM.
>
> 3. “...*claim relies on a single observation in MWS*…” : The partial solution claim is not based on a single task, and we have indeed included evidence for 4 more algorithmic tasks in the appendix (Table 1 + Appendices B.1-B.6), also pointed out in Lines 136-138. Other recent work has also shown similar results for partial solutions in n-gram estimation tasks with Transformers [Varre 25], corroborating our findings about partial solutions during Transformer training.
>
>     To further illustrate this point, we train the Transformer on concatenated sequences of increasing complexity, as below,
>
>     $$x_1, x_2, \ldots, x_{16}, {\rm SEP}, y_1, y_2, \ldots, y_{13}$$
>
>     $$y_i = \begin{cases} x_i & i\in [1, 4] \\\\ (x_i+x_{i+1}) \mod 17 & i\in [5, 9] \\\\  (x_{i+1}+x_{i+2}+x_{i+3}) \mod 17 & i\in [10, 13] \\\\ \end{cases}$$
>
>     I.e. a concatenation of Copy, MWS over 2 variables (MWS2) and MWS over 3 variables (MWS3). We find that there are separate plateaus during training, each corresponding to learning a part of the above concatenated sequences (similar to Figure 1 in [Abbe 23], and Figure 1 in [Varre 25]) – highlighting partial solutions based on “difficulty” of the respective sub-sequence.
>
> 4. We used the first 100 questions of the ARC-easy test dataset, and the last layer hidden states for all models. Further, we also measured the repetition frequency ($\rho$) of Pythia outputs, and found that it follows similar trends as cosine similarity (like the MWS case). Concretely, the table below gives the evolution of repetition frequency during the first 10000 training steps, across the 4 Pythia models discussed in the submission.
>
> |      |   0  |  1   |  2   | 4   | 8    | 16    | 32   |  64  |  128   |  256 |  512  |  1000   |  2000   | 4000  | 8000  | 10000  |
> |--|--     |--     |--     |--    |--     |--     |--    |--    |--     |--  |--    |--     |--     |--   |--    |--     |
> 14M |  0.0 | 0.0 | 0.0 | 0.0 | 0.0 | 0.0 | 0.0 | 0.97 | 0.93 | 0.98 | 0.39 | 0.13 | 0.17 | 0.12 | 0.13 | 0.11  |
> 1B |  0.03 | 0.03 | 0.03 | 0.03 | 0.99 | 0.15 | 0.81 | 0.91 | 0.83 | 0.65 | 0.19 | 0.14 | 0.13 | 0.14 | 0.17 | 0.17  |
> 1.4B |  0.16 | 0.16 | 0.16 | 0.2 | 0.99 | 0.73 | 0.89 | 0.86 | 0.8 | 0.13 | 0.15 | 0.15 | 0.13 | 0.14 | 0.15 | 0.17  |
> 2.8B |  0.04 | 0.04 | 0.05 | 1.0 | 0.82 | 0.88 | 0.97 | 0.82 | 0.79 | 0.76 | 0.14 | 0.12 | 0.16 | 0.15 | 0.17 | 0.17  |
>
>
>
>
> We will add these results through a plot similar to Fig. 8 in the updated version of the manuscript.
>
> 5. While we agree that it would be ideal to have all the results in the main paper, we have prioritized discussing the main results on a single task (MWS) in the main paper for clarity, and postpone details about other tasks and ablations to the appendix.
>
> 6. We will provide code to replicate all experiments at a later stage.
>
>
>
> *Questions:*
>
> 1. We note that repetition frequency ($\rho$) is not a flawed measure, but a clearer / stronger one since it directly measures contiguous repetitions (i.e. substrings of the form v,v,v,… for some v in vocabulary) instead of non-contiguously over the full sequence. By definition, a sequence with high repetition frequency ($\rho$) would also have a correspondingly low seq. entropy. Seq. entropy was only used to show that repetition may exist in other non-obvious forms as well, and we have not ‘changed it afterwards’ in that results for the other algorithmic tasks (i.e. except prefix sum) in the appendix still use repetition frequency ($\rho$).
>
> 2. For each step in training after $t_0$, for attention map $A \in \mathbb{R}^{(L-1) \times (L-1)}$, we use the mask $M \in \mathbb{R}^{(L-1) \times (L-1)}, M_{ij} = c$ for $(i,j) \in \Omega, M_{ij} = 1$ otherwise, except for $i=1$ (since that is a partial solution and converges early in training). Then, we use the modified attention map $A \odot M$ (elementwise product) for training and inference. We will elaborate on this in the updated version of the manuscript as well.
>
> 3. We will make necessary changes in plot styling / color palette in the updated version.
>
> 4. Token ID = integer as in the sequence, and token ID for [SEP] is already clarified wherever a task is defined. We will clarify this in the experiments section as well.
>
> We hope that our response has clarified the reviewer’s concerns about our work.
>
> [Abbe 23]: Abbe et al., 2023. SGD learning on neural networks: leap complexity and saddle-to-saddle dynamics. COLT 2023.
>
> [Varre 25]: Varre et al., 2025. Learning In-context n-grams with Transformers: Sub-n-grams Are Near-Stationary Points. ICML 2025.

---

> > ### Comment · Reviewer_dDr4 · 2025-08-03
> >
> > I thank the authors for their extensive response, and I appreciate their effort to clarify my concerns about their work.
> > However, some of my concerns remain.
> >
> > - **W1**: I do not find the provided hypothesis convincing. For instance, why would a 2-layer ReLU probe perform better than the MLP+LM head in your shallow Transformer model? I am also still convinced to some extent that the paper should at least have something of either a) or b) (proposals to alleviate the identified phenomena or a deeper understanding of their emergence), as also highlighted by Reviewer u6ru.
> >
> > - **W2**: Thanks for the additional experimental results, which make my confidence in the presented results somewhat higher. I am also sorry for my wrong statement about Line 292. Regarding the last point (pre-training of LLMs), see W4.
> >
> > - **W4**: The problem that I wanted to highlight is that using 100 (random?) samples from a single dataset does not provide, in my opinion, a basis to claim that your findings effectively extend beyond the more toy-ish settings used in the main experiments.
> >
> > - **W5-W6**: Not addressed.
> >
> > - **Q1**: I see your point. Calling it a "flawed measure" was maybe a bit excessive; however, it indeed doesn't work for some of the tasks investigated in the paper (specifically, prefix sum). So, while I understand that $\rho$ is a stronger measure for the repetition bias (as it measures the number of _consecutive_ repeated tokens), I still believe that having a single measure shared across experiments would be more valuable.
> >
> > - **Q4**: I guess that does not answer my question. Do you use a custom tokenizer or a default one?

---

> > > ### Author Response · Authors · 2025-08-05
> > > **Response to Weaknesses W1,4-6**
> > >
> > > Thank you for your response! We address weaknesses (W1,4-6) below (please see next comment for questions Q1,4) –
> > >
> > > **W1**:
> > >
> > > *”...why would a 2-layer ReLU probe perform better than the MLP+LM head…”*: The purpose of this probe is to show that the hidden states after the attention block already (non-linearly) encode information about the target before the sudden drop in loss, and subsequently the MLP + LM head fits the hidden states to the target function (hence the ‘search-then-fit’ hypothesis).
> > >
> > > In this context, it is important to note that,
> > >
> > > - we train the probe to convergence for each training step, while the updates to MLP and LM head for each training step is only once,
> > > - MLP/LM head and attention learning are coupled, so the dynamics would not be comparable to training a probe on the hidden states alone.
> > >
> > > *”...do not find the hypothesis convincing…”*: In more support of this hypothesis, note that the optimal attention results in Sec 4 (line 204) show that initializing with the correct attention map (i.e. search phase already concluded) leads to rapid convergence of loss, indicating the fitting phase in ‘search-then-fit’. Whereas, having optimal MLP/LM head weights does not help, since the model still has to perform the initial search.
> > >
> > > We note that recent work [Section C.2, Panigrahi 25] have also discussed a similar search-then-fit phenomenon for Transformers in a sparse parity task (take the $d-$dimensional input vector as a length$-d$ sequence, and predict label using last token’s output) – they study how number of attention heads in the model affects the search process.
> > >
> > > We will add the above result and discussion in the updated version. We believe that rigorously explaining this phenomenon requires substantially more work, and our paper lays the groundwork for such an analysis. While this is an important direction for future research, it falls beyond the scope of this single paper.
> > >
> > > **W4**: We would like to emphasize that we did not cherry-pick datasets or examples for this result. We further verified our claims for randomly chosen 100 questions from GSM8K (main) and ARC-Challenge test datasets, and the results are similar to Fig 8 in the original submission. We will add these results in the updated version of the manuscript.
> > >
> > > Average cosine similarity values:
> > >
> > > *GSM8K*
> > > | |0|1|2|4|8|16|32|64|128|256|512|1000|2000|4000|8000|10000|
> > > |--|--|--|--|--|--|--|--|--|--|--|--|--|--|--|--|--|
> > > 14M |  0.366 | 0.366 | 0.366 | 0.366 | 0.376 | 0.382 | 0.553 | 0.955 | 0.986 | 0.986 | 0.413 | 0.348 | 0.444 | 0.418 | 0.405 | 0.402  |
> > > 1B|  0.548 | 0.548 | 0.547 | 0.544 | 0.898 | 0.924 | 0.94 | 0.896 | 0.845 | 0.724 | 0.383 | 0.323 | 0.32 | 0.379 | 0.411 | 0.418  |
> > > 1.4B |  0.623 | 0.623 | 0.624 | 0.641 | 0.942 | 0.964 | 0.895 | 0.875 | 0.853 | 0.373 | 0.327 | 0.269 | 0.299 | 0.403 | 0.461 | 0.475  |
> > > 2.8B |  0.618 | 0.618 | 0.618 | 0.912 | 0.921 | 0.982 | 0.896 | 0.882 | 0.888 | 0.827 | 0.342 | 0.306 | 0.264 | 0.33 | 0.4 | 0.418  |
> > >
> > > *ARC-Challenge*
> > > | |0|1|2|4|8|16|32|64|128|256|512|1000|2000|4000|8000|10000|
> > > |--|--|--|--|--|--|--|--|--|--|--|--|--|--|--|--|--|
> > > 14M |  0.394 | 0.394 | 0.394 | 0.394 | 0.396 | 0.409 | 0.553 | 0.938 | 0.958 | 0.974 | 0.557 | 0.371 | 0.445 | 0.43 | 0.406 | 0.406  |
> > > 1B |  0.566 | 0.566 | 0.566 | 0.565 | 0.891 | 0.924 | 0.931 | 0.859 | 0.831 | 0.715 | 0.414 | 0.33 | 0.34 | 0.397 | 0.437 | 0.448  |
> > > 1.4B |  0.656 | 0.656 | 0.657 | 0.678 | 0.936 | 0.96 | 0.871 | 0.839 | 0.853 | 0.45 | 0.344 | 0.285 | 0.306 | 0.421 | 0.478 | 0.497  |
> > > 2.8B |  0.644 | 0.644 | 0.644 | 0.906 | 0.916 | 0.977 | 0.877 | 0.861 | 0.866 | 0.79 | 0.331 | 0.307 | 0.267 | 0.328 | 0.41 | 0.427  |
> > >
> > > **W5**: Within the 9 page limit for the main paper, we have included all the results for the MWS task essential for demonstrating the main claims made in the paper, in addition to the results on LLMs. The appendix only repeats the same experiment for other tasks to demonstrate generality of our claims, and contains implementation details and additional supporting experiments / figures / ablations / related work for reproducibility and completeness. Relevant appendix sections are linked to in the main paper wherever required for reader convenience. If the reviewer could point out specific parts in the appendix that we should include in the main paper, we are happy to attempt to move them in the updated version.
> > >
> > > **W6**: We are not allowed to share links / upload supplementary material in the rebuttal phase, which is why we stated that we will release code at a later stage.
> > >
> > > [Panigrahi 25] Panigrahi et al., 2025. Progressive distillation induces an implicit curriculum, ICLR 2025.

---

> > > > ### Author Response · Authors · 2025-08-05
> > > > **Response to Questions (Q1,4)**
> > > >
> > > > We address concerns about Q1, 4 below –
> > > >
> > > >
> > > > **Q1**:  We can indeed have a single measure (seq. entropy) for all tasks, but $\rho$ is an even stronger measure that works for all but one task. Hence, we believe including this measure adds, not subtracts, value.
> > > >
> > > > Also, as noted in the initial response - having high $\rho$ automatically implies low seq. entropy. Hence, the current set of results on repetition bias with high $\rho$ already demonstrates low seq. entropy of output sequences *for all tasks*, hence providing a unifying account of repetition bias.
> > > >
> > > >
> > > > **Q4**: We elaborate on the token embedding process below (as also noted in Line 442-443, we use minGPT implementation for toy task experiments):
> > > >
> > > > Each input sequence is represented as a list of integers (i.e. values in [0, 16] for $x_i, y_i$, 17 for SEP token) – so for batch size B and sequence length L, this is a tensor of shape [B, L] (denote by variable `idx`) containing integers in the range [0,17]. The trainable token embedding layer in the model is defined as `wte = nn.Embedding(config.vocab_size, config.n_embd)` from torch.nn library, where config.vocab_size=18 and config.n_embd=256 (embedding dimension) for MWS task. To get the `n_embd`-dimensional token embeddings, the list of integers is simply used as `tok_emb = wte(idx)` which maps the [B, L]-shaped list of integers to a [B, L, config.n_embd]-shaped real valued tensor. The embedding (`wte`) itself is a [config.vocab_size, config.n_embd] shaped tensor, where the $i^{th}$ row contains the embedding $t_i \in R^{256}$ for token $i \in [0, 17]$.

---

> > > > > ### Comment · Reviewer_dDr4 · 2025-08-05
> > > > >
> > > > > Thanks for your detailed responses; I have no further questions.
> > > > >
> > > > > Overall, I believe that the detailed responses and additional experimental results significantly contributed to increasing my confidence in the results presented by the authors, and I appreciate the engagement of the authors in the rebuttal process. I will reflect this by increasing my score. However, still cannot recommend a full acceptance of the paper, as I still believe that the paper could have been made stronger by including more insights on why the observed phenomena emerge, to what extent they represent a real problem for language models, and how to alleviate them in practice.

---

### Official Review · Reviewer_ybmK · 2025-06-27

**Clarity:** 4
**Significance:** 3
**Originality:** 3
**Rating:** 5
**Confidence:** 4

**Summary:**

This work investigates what happens during the loss plateaus. They find that the model learns a partial solution that solves easier parts of the task (e.g., predicting the 1st element in the modulo cumulative task); Sec. 3. However, at the same time, there’s a repetition bias (models output the same token over and over again) and representational collapse (tokens are nearly parallel); Secs. 3 & 5. They reaffirm that the slow learning is due to the attention map learning (Sec. 4). While most experiments are done on algorithmic toy tasks, they validate their findings in early phases of LLM pretraining (Sec. 6).

**Questions:**

* Q1: Is it ensured that each sample is only seen once during training? (train loss = test loss?)

* Q2: Is the repetition bias not just a result of representational collapse? I.e., as all hidden representations of tokens become parallel (i.e., same direction), the same token will be output. Fig. 2b would be in support of this (i.e., the curves for repetition bias and representational collapse follow a similar shape).

* Q3: Why is the 1st token position not included in the averaging of $COS_{i,j}$?

**Ethical Concerns:**

["NO or VERY MINOR ethics concerns only"]

**Final Justification:**

As stated in my comment below, the authors sufficiently addressed my concerns. I've read other reviewers' comments and the authors' responses. Besides some mentioned criticisms, I remain positive of the present work. Specifically, I addresses a typically overlooked problem (early-training loss plateau) and through controlled studies identifies the core mechanisms. While I agree with reviewer dDr4 that the connection to LLMs is a bit ad-hoc and handwavy, I think it sufficiently suggests the presence of the observed phenomenon also in real LLMs. I believe the paper has also enough merit without this part and, thus, remain with my initial rating "accept".

**Limitations:**

Yes.

**Quality:**

4

**Strengths And Weaknesses:**

## Strengths

* S1: The paper addresses an often overlooked part of training: what happens during early-training loss plateaus. Especially the connection between representational collapse and the loss plateaus is a meaningful finding.

* S2: The findings are intuitive (this is a plus!) and interesting. In particular, the observation of repetition bias and the analysis of the role of attention learning are interesting (Sec. 4)

* S3: The experiments are well-motivated, designed, and sound to support the findings. They allow fine-grained analysis to clearly work out the core mechanisms and behaviors.

* S4: The analysis is interesting and a connection to LLMs is made.

* S5: The paper is well-written.

## Weaknesses

* W1: It is unclear whether the observed phenomena (repetition bias) generalize to multiple loss plateaus or if it only happens for the first plateau. This would limit the paper’s findings. (Note that the title is a more general and I’d suggest to revise it, e.g., “What Happens During the *Early-Training* Loss Plateau?” or similar.)

* W2: It remains unclear why transformers have repetition bias in the early stages of training. The authors argue that this is an “inductive bias coming from gradient-based training” (l. 146-147) but do not further elaborate this nor provide experimental evidence.
* W3: It is somewhat expected that the attention hinders the algorithmic task to work well, as it implements the core mechanism for such tasks. It’d be interesting to also consider tasks that are less reliant on attention (though, I have to admit that I could not come up with one myself that is somewhat sensible).

* W4: The attention progress measure is not always as smooth as for the MVS task (e.g., see the multi-digit addition task in Fig. 9 right).

* W5: Code is not (yet) provided.

## Comments

* C1: References are often missing the publication venue.

* C2: The attention biasing is a bit unclear and it would be good to elaborate it in a bit more detail.

* C3: Equations should be numbered.

* C4: The related work in the appendix puts those two works - https://arxiv.org/abs/2310.12956 and https://arxiv.org/abs/2309.07311 - in the category of “grokking” (delayed generalization, i.e., low train loss with late sudden jump of val/test loss). However, these works study “breakthroughs”, “phase transitions”, or “Eureka moments” (i.e., high train & val/test loss with sudden jump of train loss). Thus, it’d suggest discussing them in the main text (along with their more related work on abrupt learning).

---

> ### Author Rebuttal · Authors · 2025-07-30
>
> We thank the reviewer for their positive feedback on our work, and are happy to know they found our findings intuitive and interesting! We address the reviewer's concerns below.
>
> *Weaknesses*:
>
> 1. We only found 1 plateau in MWS (after partial solution plateau) - once it escapes the plateau, it converges to the optimal solution. Similarly, we did not observe multiple plateaus in the tasks we consider in the paper i.e. once the model exits the first significant plateau, it converges to the optimal solution.
>
>     To address this issue, we also discuss training on a target function that shows staircase-like learning with multiple loss plateaus, each plateau corresponding to learning a (progressively more ‘difficult’) part of the sequence. Specifically, sequences of the form,
>
> $$x_1, x_2, \ldots, x_{16}, {\rm SEP}, y_1, y_2, \ldots, y_{13}$$
>
> $$y_i = \begin{cases} x_i & i\in [1, 4] \\\\ (x_i+x_{i+1}) \mod 17 & i\in [5, 9] \\\\  (x_{i+1}+x_{i+2}+x_{i+3}) \mod 17 & i\in [10, 13] \\\\ \end{cases}$$
>
> I.e. a concatenation of Copy, MWS over 2 variables (MWS2) and MWS over 3 variables (MWS3). We find that there are separate plateaus during training, each corresponding to learning a part of the above concatenated sequences (similar to Figure 1 in [Abbe 23], and Figure 1 in [Varre 25]) – highlighting partial solutions based on “difficulty” of the respective sub-sequence.
>
> We found that in this case, the tokens which are not yet predicted correctly during each plateau, also show representation collapse i.e. have high pairwise cosine similarity, indicating that the observations are not limited to a single loss plateau.
>
> 2. We provide empirical justification for gradient based training by showing that at initialization, the model does not have repetition bias or representation collapse, but only appears after a few (~20-30) steps of training.  Please see Fig 2 where the cosine similarity is quite low at the initial stages and rapidly increases during the early phase of training, and similarly for repetition frequency. This highlights that these pathologies are not an artifact of random initialization, but that the training process leads to these issues in the early stages. Understanding why these issues occur at the early stage of training is an important direction for future research, as we also note in our concluding discussion.
>
> 3. Indeed, the main idea behind our work is to show dependence on attention and how it might cause a loss plateau, which is why the tasks are designed to depend crucially on learning the correct attention map. Since one of the key features of transformers is the attention mechanism that appropriately selects relevant tokens for computing the output, we focus on this aspect in our work on learning dynamics as well.
>
> 4. We agree, although the main idea is not smoothness but that there is progress before the sudden drop, evident in the increase in attention progress measure before the sudden drop in loss. We also note that for other tasks, APM increases continuously throughout training.
>
> 5. We will provide code to replicate all experiments at a later stage.
>
>
> *Comments*:
>
> 1. , 3., 4. We will make the necessary changes as suggested by the reviewer in the updated version of the manuscript.
>
> 2. For each step in training after $t_0$, for attention map $A \in \mathbb{R}^{(L-1) \times (L-1)}$, we use the mask $M \in \mathbb{R}^{(L-1) \times (L-1)}, M_{ij} = c$ for $(i,j) \in \Omega, M_{ij} = 1$ otherwise, except for $i=1$ (since that is a partial solution and converges early in training). Then, we use the modified attention map $A \odot M$ (elementwise product) for training and inference. We will elaborate on this in the updated version of the manuscript as well.
>
>
>
> *Questions*:
>
> 1. Yes, we work in an online training setup (i.e. sample a new batch of 256 samples from the distribution at every step of training), so train loss = test loss (also noted in lines 104-107).
>
> 2. We agree that there is a correlation between representation collapse and repetitions, though would also like to point out that in the multi-digit addition task (Appendix B.1), repetition frequency remains quite low (~0.5) despite the cosine similarity being ~= 1.0. Also note that the cosine similarity is close to 1.0 but not exactly 1.0, so some hidden state representations might not exactly coincide, leading to lower repetition frequency.
>
> 3. The first output token $y_1$ is the partial solution for the MWS task, and converges to the correct output early on in training. Hence, there is no representation collapse for that token (with the other output tokens $y_2, \ldots, y_n$ that are still not correctly predicted), and hence we do not use it for our hidden state analysis.
>
>
> [Abbe 23]: Abbe et al., 2023. SGD learning on neural networks: leap complexity and saddle-to-saddle dynamics. COLT 2023.
>
> [Varre 25]: Varre et al., 2025. Learning In-context n-grams with Transformers: Sub-n-grams Are Near-Stationary Points. ICML 2025.

---

> > ### Comment · Reviewer_ybmK · 2025-08-04
> >
> > I want to thank the authors’ for their responses to my and fellow reviewers’ comments. All but one of my concerns (W1, W3-W5, C1-C4, Q1-Q3) are sufficiently addressed. Regarding *W2*: This doesn’t address the core of my concern. However, I agree with the authors that this is something worth exploring in future work. Thus, I’ll downweight this point in my final judgement.
> >
> > Overall, this is a very good paper that investigates an important question. Thus, I’d remain with my initial rating of accept.

---

### Official Review · Reviewer_ARVG · 2025-06-29

**Clarity:** 4
**Significance:** 3
**Originality:** 3
**Rating:** 5
**Confidence:** 3

**Summary:**

The main text examines a shallow transformer trained on the **moving-window-sum (MWS)** algorithmic task, using this “fruit-fly” setting to probe the broader phenomenon of abrupt jumps in loss and accuracy during transformer training. (More architectural tweaks and additional algorithmic tasks are presented in the appendices.)

In MWS, a given input sequence $x_1,\dots,x_n$ must be completed by producing $y_1 = x_1$ and, for $i \ge 2$,
$y_i = (x_i + x_{i-1}) \bmod p$.

Throughout training, the authors track quantitative metrics that capture early-stage **degeneracies**—notably repetition bias and representation collapse—which coincide with the well-known loss plateau.  By systematically tracking these signals **and demonstrating that targeted manipulations of attention scores can lengthen or remove the plateau**, the work provides initial empirical evidence that slow query–key alignment acts as a causal bottleneck, offering a concrete lead for future efforts to shorten such plateaus.

**Questions:**

**Questions on APM**

1. **Generality** – Can APM be computed for the other algorithmic tasks in Table 1?
   If so, could you list the corresponding sets $\Omega$ explicitly for each task?

2. **Support vs. weights** – APM reflects only the *support* of the attention map, not its optimal weights. This makes it a rather blunt tool. (Is this why the curve in Fig. 2 (b) tops out at \~0.8?)

3. **Placement in the paper** – APM is introduced in §2, whereas the other metrics (repetition bias $\rho$, representation-collapse **COS**) appear in §3.
   Should we view APM as conceptually distinct from $\rho$ and **COS**, or would it be clearer to define all three together?

---

**Miscellaneous**

4. **Terminology — “degeneracy.”**  The term can sometimes have a precise mathematical/technical meaning, yet here it seems to label undesirable model behaviours (repetition bias, representation collapse, etc.).  Would a word like “pathology” or “failure mode” be clearer?

5. **Styling** – Loss and accuracy are plotted with different styles in Figs. 1 and 2; please make them consistent.

6. **Colour reuse** – In Fig. 2 the two panels recycle the same colours for different quantities, which can be confusing; consider distinct palettes.

**Ethical Concerns:**

["NO or VERY MINOR ethics concerns only"]

**Final Justification:**

I maintain my positive score.

**Limitations:**

Yes

**Quality:**

4

**Strengths And Weaknesses:**

**Strengths**

* **Transparent setup.** Task, metrics, and training regimen are described clearly, so the experiments are easy to replicate.
* **Causal interventions.** The paper’s *Biasing the Attention Map* and *Training with Optimal Attention* experiments in Section 4 show that directly manipulating the attention map lengthens or removes the plateau, supporting the bottleneck claim.
* **Quantitative metrics.** Repetition rate, cosine-collapse, and APM turn vague “degeneracy” into reproducible numbers.
* **Evidence is quite extensive.** The same patterns appear on several algorithmic tasks and are glimpsed in early LLM checkpoints, hinting at broader relevance.

**Weaknesses**

* The idea of a partial solution feels a bit arbitrary. A solution can be partial in myriad ways.
* The APM metric is not readily available in most tasks as we might not know the optimal $\Omega$. Furthermore, the metric just assesses whether the attention has the right support, not the optimal attention values, as far as I can tell.

---

> ### Author Rebuttal · Authors · 2025-07-30
>
> We thank the reviewer for their positive evaluation of our work! We address their concerns (about partial solution, attention progress measure, etc.) below.
>
> *Weaknesses*:
>
> 1. While we agree that a partial solution may not be unique for a given task (i.e. multiple partial solutions could be defined for the same task), we have focused on partial solutions that occur consistently during training in our setup, and also found that they can be understood intuitively w.r.t. the original task. In other words, a partial solution is simply the capability the model has already learned before the sudden drop in loss.
>
> 2. “...*APM metric is not readily available in most tasks*…”: We provide the optimal $\Omega$ for all tasks considered in our paper below.
>
>     Here we assume the attention map is of the shape $A \in R^{(L-1) \times (L-1)}$ in the next token prediction setup on the full input sequence. Hence, for {MWS, Prefix sum, Histogram, Reverse, Copy}, $L=33$, for permutation $L=50$, and for Multi-digit addition $L=15$. Further, the $(i, j)$ below are in the set $[L-1] \times [L-1]$ to denote row / column of $A$.
>
>
>
> | Algorithmic Task	 | APM set $(\Omega \subset [L-1] \times [L-1])$ |
> |--------------------|----------|
> | MWS    		 | $\\{(16,0)\\} \cup \\{(16+i, i-1) \mid i\in[1,15]\\} \cup \\{(16+i, i) \mid i\in[1,15]\\}$|
> | Prefix Sum    		 | $\\{(16,0)\\} \cup \\{(16+i, 16+i) \mid i\in[1,15]\\} \cup \\{(16+i, i) \mid i\in[1,15]\\}$|
> | Multi-digit Addition   		 | $\\{(i, 12-i) \mid i\in[9,12]\\} \cup \\{(i, 17-i) \mid i\in[9,12]\\} \cup \\{(i, 13-i) \mid i\in[10,13]\\} \cup \\{(i, 18-i) \mid i\in[10,13]\\}$|
> | Permutation | Layer 1: $\\{(17+i, \pi_{i+1}-1) \mid i\in[0,15]\\}$|
> || Layer 2: $\\{(33+i, 17+i) \mid i\in[0,15]\\}$|
> | Histogram | Layer 1: $\\{(16+i, i) \mid i\in[0,15]\\}$|
> | Copy | $\\{(16+i, i) \mid i\in[0,15]\\}$|
> | Reverse | $\\{(16+i, 15-i) \mid i\in[0,15]\\}$|
>
>
>
> - “...*just assesses whether the attention has the right support, not the optimal attention values*…”: We note that APM is intended to measure attention score magnitude at the correct (support) token positions relative to other positions, and can be considered a proxy to measuring attention probabilities in softmax Attention. The overall idea is that the model should attend to the correct tokens in the sequence, and not the other (irrelevant) tokens, and this requires that the attention score has higher magnitude at the support positions relative to other tokens. We note that for MWS the APM reaches 0.8 highlighting that the attention map attends to the correct tokens with ~80% of available attention scores.
>
>
> We further note that the optimal attention map in our tasks is usually sparse, and hence measuring its support is already very informative of the progress. Moreover, the optimal attention score values may not be unique, so measuring the magnitude of these values without taking into account the support would not be too useful.
>
> *Questions on APM:*
>
> 1. Yes indeed, we have reported APM dynamics for all tasks, as well as the respective attention heatmaps in Appendix B.1-B.6. We have summarized APM sets $\Omega$ for all tasks in response to Weaknesses.
>
> 2. Please see our response to weakness regarding APM above.
>
> 3. We note that APM is more a property of the attention map (hence model architecture), which is why we chose to define it with the architecture setup for clarity. COS is for hidden states and $\rho$ for output token sequences, and we chose to define them with the associated collapse / repetition phenomena. A conceptual distinction was not intended, since all these quantities are ultimately used for understanding the training dynamics.
>
> Miscellaneous:
>
> 4. We agree and will replace degeneracy by a more suitable word as suggested.
> 5. and 6. We will make the required changes in plot styling / color palette in the updated version.
>
> We hope our response has addressed reviewer concerns.

---

> > ### Comment · Reviewer_ARVG · 2025-08-08
> > **thank you for your response**
> >
> > Thank you for the thorough rebuttal and for addressing my earlier comments. I continue to think this is a well-executed and worthwhile paper, and my score of 5 remains unchanged.
> >
> > In reading the other reviews, I note a common thread: while the empirical characterisation of phenomena like repetition bias, representation collapse, and slow attention learning is valuable, there is still no fully developed mechanistic or theoretical account of why these arise, nor strong evidence that the observed dynamics generalise broadly beyond the studied settings. I agree this is a natural direction for future work, and I appreciate that the authors have acknowledged these limits.
> >
> > Overall, I believe the contribution justifies acceptance.

---

### Official Review · Reviewer_u6ru · 2025-07-01

**Clarity:** 3
**Significance:** 3
**Originality:** 3
**Rating:** 4
**Confidence:** 4

**Summary:**

This paper studies the learning dynamics of Transformers on algorithmic tasks, focusing on the common pattern of a long performance plateau followed by a sudden improvement. The authors find that during the plateau, models develop partial solutions but suffer from repetition bias and representation collapse. These issues stem from slow learning of attention patterns, which, once resolved, lead to rapid gains in performance. The paper also shows that similar phenomena appear in early training of LLMs like Pythia and OLMo, highlighting their broader relevance.

**Questions:**

1. Can you provide any hypotheses or theoretical intuitions for why attention learning is slow or why representation collapse occurs during the plateau phase?

2. How sensitive are the observed dynamics to initialization and optimizer choice? It would be helpful to know whether these behaviors are optimizer- or hyperparameter-specific, or if they robustly generalize across settings.

**Ethical Concerns:**

["NO or VERY MINOR ethics concerns only"]

**Final Justification:**

While the paper compellingly identifies important phenomena such as repetition bias, representation collapse, and attention as a potential bottleneck, it stops short of explaining why this bottleneck arises. I understand the authors acknowledge this as future work and provided some hypotheses in the rebuttal upon my request, which I appreciate. However, the lack of theoretical framing or concrete mechanistic explanation in the main paper limits the depth of the current contribution. The work would be stronger with a simple toy model or analysis to support these observations. That said, I find the empirical findings interesting and valuable, and I am raising my score to a weak acceptance.

**Limitations:**

Yes

**Quality:**

2

**Strengths And Weaknesses:**

**Strengths**

1. The paper is very well written, with good organization and clarity.

2. The characterization of repetition bias, representation collapse, and hidden progress in attention learning provides a useful lens through which to understand transformer training dynamics.

**Weaknesses**

1. While the paper compellingly identifies phenomena like repetition bias, representation collapse, and attention as the underlying bottleneck, it doesn't go deeper into why this bottleneck arises. I understand this is noted by the authors as a future direction, but without at least some theoretical framing or hypothesis testing, the insight feels limited. The paper would be stronger with a toy model or a mechanistic explanation.

2. Some of the findings feel like they might be sensitive to training details. What happens if you change the optimizer, add regularization, or tweak the learning rate schedule? Right now it’s hard to tell whether the observed effects are fundamental or just artifacts of a particular training setup. A few ablations on this would really help clarify robustness. I’d be open to raising my score if this point is addressed convincingly in the rebuttal.

- A good example see When Attention Sink Emerges in Language Models: An Empirical View https://arxiv.org/abs/2410.10781.

---

> ### Author Rebuttal · Authors · 2025-07-30
>
> We thank the reviewer for their encouraging comments, and are happy to know they found our work well written! We address the reviewer’s concerns (about hypotheses for observed phenomena, robustness to hyperparameters, etc.) below.
>
>
> *Weaknesses*:
>
> 1. **Hypotheses for observed phenomena**
>
> *Slow learning of attention map*:
>
> Our hypothesis for loss plateau is that the attention layer first searches and combines the correct input tokens, and subsequently during training the MLP layer and LM head learn to map those hidden states to the correct output tokens. This is verified via non-linear (2-layer ReLU MLP) probing the outputs of Attention block during various points in training, and we find that such a probe is able to map the hidden states to the correct output labels earlier in training than the final model output (more details below). We note that this is related to prior work on the “search-then-fit” mechanism in 2–layer neural nets [Abbe 23], where the first layer in the neural network searches for the support variables of the target function in the input vector, and subsequently the 2nd layer fits the target function using these support variables.
>
> [Probing details]: We probe hidden state outputs $h_i$ of the Attention block corresponding to the output tokens, before they are added to the residual stream. We map these hidden states to the correct output tokens (i.e. $y_i$ as defined for MWS) using a 2-layer ReLU MLP probe ${R}^{256} \ni x  \mapsto W_2 \sigma(W_1 x) \in {R}^{17}, W_1 \in R^{256 \times 256}, W_2 \in R^{17 \times 256}$ mapping to logits over relevant vocabulary {0,...,16} (no SEP=17 token since that is not relevant for $y_i$). We implement the probe using MLPClassifier from the sklearn.neural_network library, train it using 1024 samples, and evaluate its performance using a held-out test set of 1024 samples.
>
> We will add this experiment to the updated version of the paper; a deeper theoretical understanding of this phenomenon is an important direction for future research.
>
> *Representation collapse*: We hypothesize that representation collapse arises from a bias of Transformers towards repetitive outputs in autoregressive setup; towards this hypothesis, we showed that training on repetitive sequences (hence explicitly encouraging representation collapse / repetition bias) is easier in the sense of training loss decreasing rapidly and monotonically, without any loss plateaus (Sec 5.2).
>
>
> 2. **Robustness to hyperparameters**: We show that abrupt learning occurs across a range of optimization hyperparameter choices as below.
>
> To summarize, results for all hyperparameter and optimizer choices have abrupt learning, representation collapse and repetitions, with varying length of plateau / extent of representation collapse depending on learning rate, batch size, initialization scale, choice of optimizer. Note that for all ablations, the other hyperparameters remain fixed to default values as described in the paper.
>
> - *Optimizer*: We have already reported results on SGD (that are similar to the Adam case) in our submission (Fig. 26 in Appendix E). In addition, we experimented with
>
> a) AdamW (lr=1e-4, wd={1e-3, 1e-2, 5e-2, 1e-1}) and
>
> b) Muon (lr=0.02) with {Adam (lr=1e-4), AdamW (lr=1e-4, wd=1e-2)} for embedding, LM head and all bias parameters.
>
> and found that the results are similar to those with Adam (abrupt learning, representation collapse and repetitions occur), although Muon appears to have a shorter loss plateau.
>
> - *Learning rate schedule*: We experimented with {linear warmup, linear warmup with cosine decay}, with warmup steps {20, 50, 100, 200} for a total 400 training steps and did not find any significant changes in training characteristics. (Note that cosine decay leads to 0 learning rate at the end of training). We also note that prior work [Chen 24] used linear warmup in their training setup and observed abrupt learning.
>
> Additionally, we also tested the learning rate schedule in the suggested paper on attention sink with some modifications to total steps and warmup steps for our setup (i.e., total steps=400, warmup steps=20, rest of the hyperparams like AdamW weight decay, min / max learning rate, beta1, beta2, gradient clipping remain the same as the suggested paper).
>
> - *L2 Regularization*: We experimented with Adam with weight decay = {1e-3, 1e-2, 5e-2}, and found that while 1e-3, 1e-2 cases are similar to the unregularized training results, for wd=5e-2, the plateau is slightly longer while the peak cosine similarity is slightly lower (~0.8)
>
> - *Adam + constant LR*: We experimented with learning rate = {5e-7, 1e-6, 1e-5, 1e-3} and did not find significant changes in training characteristics except that with smaller learning rates, the plateau is longer as expected.
>
> - *Batch size*: We vary batch size = {8, 32, 1024, 4096} and did not find changes in training characteristics except that with smaller batch sizes, the plateau is longer as expected.
>
> - *Initialization scale*: Default implementation initializes entries of all linear / embedding layers i.i.d. from $\mathcal{N}(0, 0.02^2)$ except for entries of $W_O, W_2$ matrices initialized i.i.d. from $\mathcal{N}(0, (0.02^2)/2L)$, for $L$ layer model ($L=1$ in our case). We vary the scale of  initialization by replacing $0.02$ by $0.02 \alpha$ for $\alpha \in$ {1e-3, 1e-2, 1e-1, 1e1}, and find that for $\alpha < 1$, the loss plateau is longer and the cosine similarity goes to $\approx 1.0$ faster. Whereas for $\alpha = 10$, the cosine similarity peaks at around $0.6$ and the loss plateau is longer than $\alpha=1$ case.
>
> | Sweep         		 | Value            		   | Abrupt Learning and RC |
> |------------------------|-----------------------------|------------------------|
> | Optimizer    			 | AdamW 			 		   | Yes, similar 					|
> |         	   			 | Muon + Adam     	 		   | Yes, shorter plateau 	|
> |              			 | Muon + AdamW     		   | Yes, shorter plateau 	|
> | Learning rate schedule | Linear Warmup 			   | Yes, similar 					|
> |  						 | Linear Warmup + Cosine Decay| Yes, similar 					|
> |  						 | [Gu 25] (suggested by reviewer, adapted to our task)       			   | Yes, similar 					|
> | L2 regularization 	 | 1e-3, 1e-2 				   | Yes, similar           			|
> |  						 | 5e-2       				   | Yes, slightly longer plateau, cosine similarity peak ~0.8 |
> | Learning Rate		 	 | 5e-7, 1e-6, 1e-5 		   | Yes, longer plateau  			|
> |  						 | 1e-3       				   | Yes, similar           			|
> | Batch Size		 	 | 8, 32 					   | Yes, longer plateau  	|
> |  						 | 2048, 4096       		   | Yes, similar           			|
> | Initialization Scale	 | 1e-3, 1e-2, 1e-1			   | Yes, longer plateau, stronger RC |
> |  						 | 1e1       		   		   | Yes, longer plateau, weaker RC (peak cosine sim ~0.6) |
>
> (RC= Representation collapse)
>
> *Questions*:
>
> 1. Please see response to Weakness 1 above.
>
> 2.
>
> - Hyperparameter / optimizer choice: please see response to Weakness 2.
>
>
> - Initialization choice: We note that we do not set random seed for any experiment, hence this training behavior is not limited to a specific random initialization. The behavior however is sensitive to initialization scale i.e. variance of the normal distribution used to sample initial weights (as expected based on prior work in optimization theory for neural nets); please see initialization scale results in response to Weakness 2 above.
>
> We hope that our response has resolved the reviewer’s concerns about the robustness of our results to hyperparameter and optimizer choices, and that our hypothesis about why abrupt learning occurs (supported by preliminary experimental results) has helped towards gaining a theoretical / mechanistic intuition for this phenomenon.
>
>
> [Abbe 23]: Abbe et al., 2023. SGD learning on neural networks: leap complexity and saddle-to-saddle dynamics. COLT 2023.
>
> [Chen 24]: Chen et al., 2024. Sudden Drops in the Loss: Syntax Acquisition, Phase Transitions, and Simplicity Bias in MLMs. ICLR 2024.
>
> [Gu 25] Gu et al., 2025. When Attention Sink Emerges in Language Models: An Empirical View. ICLR 2025.
>
> [Loshchilov 19]: Loshchilov et al. 2019. Decoupled Weight Decay Regularization. ICLR 2019

---

> > ### Comment · Reviewer_u6ru · 2025-08-06
> >
> > Thank you for the detailed response. I am happy to see the thorough new ablation results validating the robustness of the observation. I will raise my score to 4.

---

### Decision · Program_Chairs · 2025-09-17

**Decision:**

Accept (poster)

**Comment:**

This paper studies the training dynamics of transformers for simple algorithmic tasks.
In particular, the focus is about what happens during the long loss plateau phase of training, which is often followed by a sudden drop in loss.
It has been shown that during the loss plateau, the model will learn partial solutions while suffering from repetition bias and representation collapse, due to slow key-query alignment.
These phenomena can also be found in early training of large language models.

Overall the paper is well-written, and the authors have included extensive experimental details to illustrate the loss plateau phase.
The findings are valuable for further understanding of the training dynamics of transformers.
However, as noted by the reviewers, the current paper falls short in the investigation of the mechanisms behind the observed phenomena.
Nonetheless, the empirical observations would provide insights for future work.

For paper revision, the authors should include ablation studies with respect to the training configurations as provided during rebuttal.
Other comments and discussions should also be incorporated.